# DRIFT: Decoupled Rollouts and Importance-Weighted Fine-Tuning for Efficient Multi-Turn Optimization

**Jian Mu**[1]  **Tianyi Lin**[1]  **Chengwei Qin**[1]  **Zhongxiang Dai**[2]  **Yao Shu**[1]

## Abstract

Large language models are increasingly deployed in multi-turn interactive settings where users or environments can iteratively provide lightweight feedback. Unfortunately, optimizing such behavior presents a sharp dilemma in practice: online reinforcement learning is able to effectively address multi-turn dynamics but is prohibitively expensive due to the cost of generating full correction trajectories at every update, whereas offline supervised fine-tuning (SFT) is efficient but suffers from distribution shift and behavioral collapse. To this end, we novelly propose DRIFT (*Decoupled Rollouts and Importance-Weighted Fine-Tuning*), a framework that operationalizes the theoretical insight that the KL-regularized RL objective is equivalent to importance-weighted supervised learning. DRIFT decouples rollout from optimization by sampling offline interaction trajectories from a fixed reference policy, deriving return-based importance weights, and optimizing the policy via weighted SFT on the resulting dataset. Empirically, we demonstrate that DRIFT matches or exceeds the performance of multi-turn reinforcement learning baselines while maintaining the training efficiency and simplicity of standard supervised fine-tuning. Code is available at https://github.com/2020-qqtcg/DRIFT.

## 1. Introduction

Large language models (LLMs) (Guo et al., 2025; Qwen et al., 2025; Comanici et al., 2025) have evolved from static query-response engines into interactive agents capable of advanced reasoning. While standard training pipelines fo-

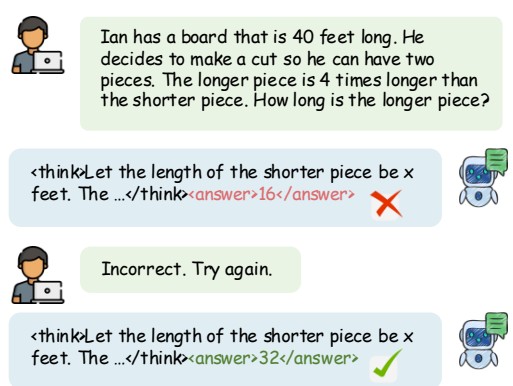

*Figure 1.* Multi-turn interaction. The user engages in a dialogue with the LLM. If the LLM provides an incorrect response, the user offers simple feedback to point out the error. The LLM then re-attempts the task until a correct answer is generated or the maximum number of turns is reached.

cus predominantly on single-turn accuracy (Rafailov et al., 2023; Bai et al., 2025; Zheng et al., 2025b), real-world deployment necessitates multi-turn capabilities where users iteratively provide feedback to guide the model (Li et al., 2025; Laban et al., 2025). However, models trained strictly on single-turn data often exhibit fragility when confronted with negative feedback (Gao et al., 2024; Kumar et al., 2024; Zhang et al., 2025; Liu et al., 2025), frequently repeating errors or degrading in performance after a revision attempt. As shown in Figure 1, effectively leveraging lightweight feedback signals (e.g., "Incorrect, please try again") to improve robust multi-turn reasoning remains a critical open challenge.

Current approaches to multi-turn optimization face a sharp dilemma between effectiveness and efficiency. On one hand, Supervised Fine-Tuning (SFT) on offline correction trajectories is sample-efficient but often fails to learn genuine correction policies. As noted by Kumar et al. (2024), naive SFT suffers from distribution shift and behavioral collapse, where models over-optimize for first-turn accuracy while failing to produce meaningful edits in subsequent turns. On the other hand, Online Reinforcement Learning (RL) (Gao et al., 2024; Kumar et al., 2024; Liu et al., 2025) approaches like PPO (Schulman et al., 2017; Ouyang et al., 2022) or

---
[1]The Hong Kong University of Science and Technology (Guangzhou) [2]The Chinese University of Hong Kong, Shenzhen. Correspondence to: Yao Shu < yaoshu@hkust-gz.edu.cn>.

*Proceedings of the 43rd International Conference on Machine Learning*, Seoul, South Korea. PMLR 306, 2026. Copyright 2026 by the author(s).

GRPO (Shao et al., 2024; Guo et al., 2025) address these distribution issues but incur a prohibitive computational cost. Unlike single-turn settings, multi-turn optimization requires generating full interaction trajectories for every policy update. As illustrated in Figure 6, this rollout cost scales poorly with interaction length, making standard online RL hard for training on multi-turn reasoning tasks.

We address this bottleneck by establishing a fundamental connection between the KL-regularized RL objective and weighted supervised learning. Our core insight is that the gradient of the online RL objective can be approximated using offline trajectories sampled from a fixed reference policy, provided they are re-weighted by their exponentiated rewards. This theoretical equivalence implies that the expensive rollout generation can be completely *decoupled* from the policy optimization process. By shifting the computational burden to an offline, parallelizable generation phase, we can project the benefits of online RL into a high-throughput supervised training framework without the complexity of online interactions.

Building on this insight, we propose DRIFT (*Decoupled Rollouts and Importance-Weighted Fine-Tuning*), a practical framework for verifiable multi-turn optimization. DRIFT operates in two distinct stages: (1) an offline rollout stage where interaction trajectories are collected under a reference policy, and (2) a trajectory-weighted SFT stage where the model is optimized using our derived importance weights. This design enables DRIFT to match the asymptotic performance of online RL baselines while retaining the computational efficiency of standard SFT.

Our contributions are summarized as follows:

- We propose DRIFT, a method that decouples rollout from optimization to realize an RL objective via weighted SFT, thereby enabling efficient training under multi-turn interaction protocols.

- We provide theoretical results demonstrating that our weighted objective is equivalent to the underlying KL-regularized RL objective, ensuring the effectiveness and stability of the optimization.

- Extensive experiments show that DRIFT achieves performance comparable to or better than online RL baselines across mathematical and general reasoning benchmarks while offering substantially higher training efficiency.

## 2. Problem Setup

We formulate the multi-turn answer correction task as a finite-horizon Markov Decision Process (MDP), defined by the tuple $\mathcal{M} = (\mathcal{S}, \mathcal{A}, \mathcal{P}, \mathcal{R}, \gamma, T)$, where $T$ represents the maximum turn budget. We detail the components and the underlying assumptions below.

**States and Actions.** Let $\mathcal{V}$ denote the vocabulary. The action space $\mathcal{A} = \mathcal{V}^*$ consists of sequences of tokens generated by the model. At turn $t$, the action $y_t \in \mathcal{A}$ represents the model's response. The state space $\mathcal{S}$ represents the interaction history. Given an initial problem prompt $x_1$, the state at turn $t$ is defined as the full interaction history up to that point: $x_t = (x_1, y_1, f_1, \ldots, y_{t-1}, f_{t-1})$, where $f_i$ denotes the feedback received at turn $i$. Since Large Language Models are autoregressive and condition on the entire input context, treating the complete interaction history $x_t$ as the state ensures that the Markov property holds strictly, as $x_t$ contains all sufficient statistics for the next transition.

**Transition Dynamics.** The transition function $\mathcal{P} : \mathcal{S} \times \mathcal{A} \to \mathcal{S}$ is deterministic in our setting. A verifier function $\mathcal{V}(y_t)$ evaluates the correctness of the response $y_t$. If $y_t$ is correct, the episode terminates successfully. If $y_t$ is incorrect and $t < T$, the environment transitions to $x_{t+1}$ by appending a lightweight feedback message $f$ (e.g., "Incorrect, please try again"):

$$x_{t+1} = \text{concat}(x_t, y_t, f). \tag{1}$$

We assume a deterministic transition dynamics with fixed feedback $f$. This assumption aligns with standard evaluation protocols in reasoning benchmarks (Liu et al., 2025; Kumar et al., 2024), where the user provides consistent, rigorous feedback to elicit self-correction without introducing stochastic user noise.

**Objective.** Let $\pi_\theta(y|x)$ be the parameterized policy initialized from a reference model $\pi_{\text{ref}}$. A trajectory is a sequence $\tau = (x_1, y_1, \ldots, x_L, y_L)$, where $L \leq T$ is the effective episode length. We define a discounted trajectory return:

$$R(\tau) \triangleq \sum_{t=1}^{L} \gamma^{t-1} r(x_t, y_t), \quad \gamma \in (0, 1), \tag{2}$$

where $r(x_t, y_t) = 1$ if $y_t$ is correct and 0 otherwise. The discount factor $\gamma$ strictly penalizes delayed success, incentivizing the model to correct errors as early as possible. Our goal is to optimize the standard KL-regularized RL objective (Ouyang et al., 2022; Rafailov et al., 2023), which balances maximizing the expected return against maintaining fidelity to the reference policy:

$$\begin{aligned} \max_\theta \ J(\theta) &\triangleq \mathbb{E}_{\tau \sim p_\theta}[R(\tau)] \\ &\quad - \beta \, \text{KL}(p_\theta(\cdot \mid x) \,\|\, p_{\text{ref}}(\cdot \mid x)). \end{aligned} \tag{3}$$

where $\beta > 0$ controls the strength of the regularization, and $p_\theta(\tau \mid x)$ and $p_{\text{ref}}(\tau \mid x)$ are the trajectory distributions defined by $\pi_\theta$ and $\pi_{\text{ref}}$, respectively. This formulation grounds our multi-turn correction problem as a specific instance of

return maximization under distribution constraints. However, optimizing Eq. (3) with standard online RL requires generating fresh on-policy multi-turn trajectories from $p_\theta$ at every update, so the rollout cost scales with the interaction horizon.

## 3. From KL-Regularized RL to Weighted SFT

In this section, we establish a fundamental equivalence between the online KL-regularized reinforcement learning objective and importance-weighted supervised fine-tuning.

### 3.1. The Optimal Trajectory Distribution

To manipulate this objective effectively, it is instructive to first abstract away the parametric constraints and identify the theoretically optimal distribution $p^\star(\cdot|x)$ that maximizes Eq. (3) over the space of all valid probability distributions.

**Theorem 1** (Optimal Trajectory Distribution). *For a fixed prompt $x$ and bounded return $R(\tau)$, the variational problem corresponding to the RL objective admits a unique closed-form maximizer $p^\star(\tau \mid x)$, defined as an exponential tilting of the reference distribution:*

$$p^\star(\tau \mid x) = \frac{1}{Z(x)} \, p_{\text{ref}}(\tau \mid x) \, \exp\!\left(\frac{R(\tau)}{\beta}\right), \quad (4)$$

*where $Z(x) \triangleq \mathbb{E}_{\tau \sim p_{\text{ref}}(\cdot|x)}[\exp(R(\tau)/\beta)]$ is the prompt-dependent partition function.*

This result is pivotal because it reveals that the ideal correction behavior is not arbitrary; it essentially re-weights the reference behavior such that trajectories with higher returns $R(\tau)$ are assigned exponentially higher probability mass. The parameter $\beta$ serves as the temperature of this distribution, controlling the sharpness of the tilt towards high-return regions.

### 3.2. Connecting RL to Distribution Matching

Having identified the theoretically optimal target $p^\star$, we now establish the fundamental connection between the standard reinforcement learning objective and distribution matching. While $p^\star$ was derived as the maximizer of the variational problem, it is not immediately obvious how the parametric RL objective $J(\theta)$ relates to this distribution. The following theorem bridges this gap by explicitly reframing the reward maximization problem as a divergence minimization task.

**Theorem 2** (RL as Reverse-KL Minimization). *Let $J(\theta)$ denote the KL-regularized objective defined in Eq. (3), and let $Z(x)$ be the partition function independent of $\theta$. The RL objective satisfies the following identity:*

$$J(\theta) = \beta \log Z(x) - \beta \, \text{KL}(p_\theta(\cdot|x) \,\|\, p^\star(\cdot|x)). \quad (5)$$

*Since $\beta \log Z(x)$ is constant with respect to $\theta$, maximizing the expected return is mathematically equivalent to minimizing the Reverse-KL divergence between the policy $p_\theta$ and the optimal distribution $p^\star$:*

$$\arg\max_\theta J(\theta) = \arg\min_\theta \text{KL}\left(p_\theta(\cdot \mid x) \,\|\, p^\star(\cdot \mid x)\right). \quad (6)$$

Theorem 2 is significant because it explicitly characterizes the ideal optimization path: the policy $p_\theta$ should strive to cover the mode of $p^\star$. Standard online RL algorithms, such as PPO, effectively minimize this Reverse-KL divergence. However, a critical bottleneck arises from the direction of the divergence. The term $\text{KL}(p_\theta\|p^\star)$ involves an expectation under the *current* policy $p_\theta$ (i.e., $\mathbb{E}_{\tau \sim p_\theta}[\dots]$). Estimating its gradient requires generating fresh rollouts from $p_\theta$ at every optimization step, incurring the high computational costs we aim to avoid.

To overcome this, we propose utilizing the *Forward-KL divergence*, $\text{KL}(p^\star \,\|\, p_\theta)$, as a surrogate objective. This substitution allows us to shift the expectation from the changing policy $p_\theta$ to the fixed optimal distribution $p^\star$. We formalize this substitution in two complementary ways: an exact global statement under realizability, and a local surrogate guarantee that does not require realizability.

**Lemma 3** (Forward/Reverse KL Duality). *Assume the policy class $\Pi_\theta$ is sufficiently expressive to contain the optimal distribution (i.e., $\exists\theta^\star$ such that $p_{\theta\star} = p^\star$). Under this realizability assumption, the set of global minimizers for the Forward-KL and Reverse-KL divergences coincide:*

$$\begin{aligned}
\{\theta : p_\theta = p^\star\} &= \arg\min_\theta \text{KL}(p_\theta \,\|\, p^\star) \\
&= \arg\min_\theta \text{KL}(p^\star \,\|\, p_\theta).
\end{aligned} \quad (7)$$

Lemma 3 gives an exact global guarantee, but its role is limited to the realizable case: it uses $p^\star \in \Pi_\theta$ only to identify the global minimizers of the two KL directions. In finite-capacity models, $p^\star$ may not be exactly realizable, and the global projections induced by Forward-KL and Reverse-KL need not coincide. We therefore complement Lemma 3 with a local comparison result showing that the two KL objectives still have the same second-order geometry around $p^\star$.

**Lemma 4** (Local validity without realizability). *Fix a prompt $x$, and write*

$$P^\star = p^\star(\cdot \mid x), \qquad P_\theta = p_\theta(\cdot \mid x).$$

*Assume that $P^\star$ has a finite effective support. Then there exist constants $\varepsilon_x > 0$ and $C_x < \infty$ such that, for any $P_\theta$ on the same support with*

$$\text{TV}(P_\theta, P^\star) \le \varepsilon_x,$$

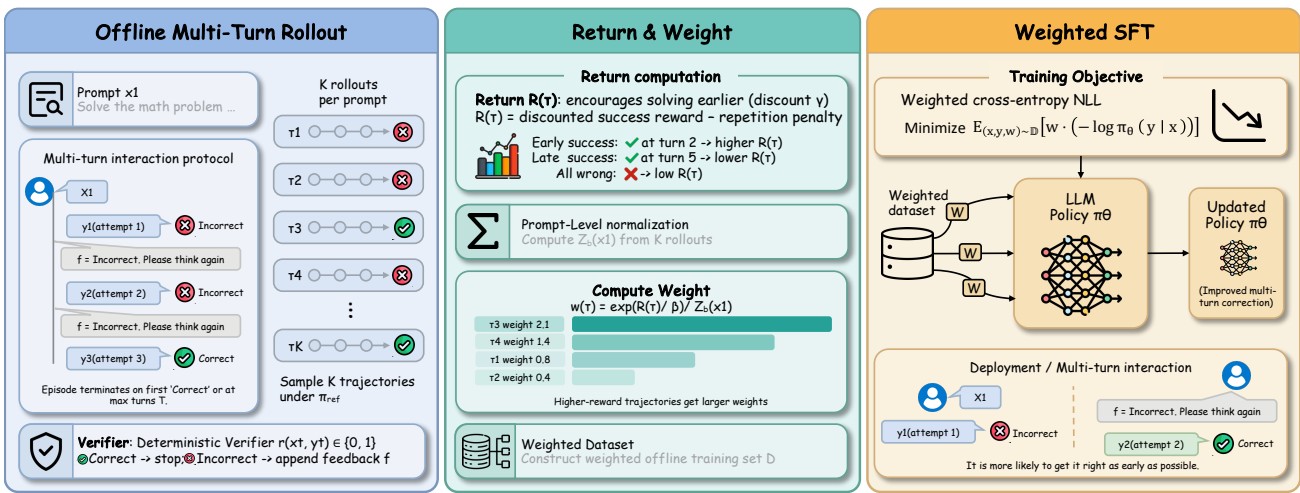

*Figure 2.* DRIFT overall framework overview. DRIFT consists of two stages: (1) an offline rollout stage, where a batch of trajectories is sampled once from the reference policy and trajectory weights are computed based on the return; and (2) a weighted supervised optimization stage, where the collected $(x, y, w)$ tuples are used for weighted SFT. This fully decouples rollout from training, enabling DRIFT to achieve RL objectives with efficiency close to standard SFT.

*we have*

$$|\mathrm{KL}(P_\theta \| P^\star) - \mathrm{KL}(P^\star \| P_\theta)| \leq C_x \, \mathrm{TV}(P_\theta, P^\star)^3.$$

*Thus, even without realizability, Forward-KL and Reverse-KL share the same local second-order geometry around $p^\star$.*

Together, Lemmas 3 and 4 justify the use of Forward-KL as a training surrogate at two levels. In the realizable case, the substitution is exact at the level of global optima; in the non-realizable case, it remains locally faithful whenever $p_\theta$ is close to $p^\star$. We now exploit the computational advantage of the Forward-KL objective: its only $\theta$-dependent term is the cross-entropy under the fixed target distribution $p^\star$. Hence,

$$\min_\theta \mathrm{KL}(p^\star \| p_\theta) \iff \max_\theta \mathbb{E}_{\tau \sim p^\star(\cdot|x)}[\log p_\theta(\tau|x)]. \quad (8)$$

### 3.3. Deriving the Offline Weighted Objective

The challenge remains that we cannot sample directly from the optimal distribution $p^\star$ to compute this expectation. However, since $p^\star$ is absolutely continuous with respect to the reference policy $p_{\mathrm{ref}}$, we can apply importance sampling to change the measure from $p^\star$ to $p_{\mathrm{ref}}$. This transformation is the core engine of DRIFT.

**Theorem 5** (Equivalence to Importance-Weighted SFT).
*By rewriting the expectation over $p^\star$ as an expectation over $p_{\mathrm{ref}}$ weighted by $w(\tau|x) = p^\star(\tau|x)/p_{\mathrm{ref}}(\tau|x)$, we obtain the following tractable objective:*

$$\mathcal{L}(\theta) \triangleq \mathbb{E}_{\tau \sim p_{\mathrm{ref}}(\cdot|x)}\left[-w(\tau|x) \log p_\theta(\tau|x)\right], \quad (9)$$

*where the importance weight for each trajectory is given by:*

$$w(\tau|x) = \frac{\exp\left(R(\tau)/\beta\right)}{Z(x)}. \quad (10)$$

This formulation fully decouples the rollout phase from the optimization phase. We can effectively approximate $\mathbb{E}_{\tau \sim p_{\mathrm{ref}}}$ using a static dataset of trajectories sampled offline from the reference policy. The learning process then reduces to a standard supervised fine-tuning loop where each trajectory loss is scaled by its normalized exponential return. For autoregressive language models, the term $\log p_\theta(\tau|x)$ naturally decomposes into a sum of token-level log-probabilities, allowing Eq. (5) to be implemented by simply applying the scalar weight $w(\tau|x)$ to the cross-entropy loss of every token in the response.

## 4. The DRIFT Algorithm

### 4.1. Overview of DRIFT

The theoretical analysis in Section 3 establishes that the complex online RL objective can be equivalently transformed into a tractable weighted supervised learning problem. Guided by this theoretical equivalence, we propose DRIFT, a framework that operationalizes this insight to fundamentally decouple interaction rollouts from policy optimization. As illustrated in Figure 2, DRIFT effectively bridges the gap between the rigorous RL objective and practical supervised fine-tuning through two distinct stages:

**Stage 1: Offline Trajectory Generation.** Instead of expensive online interactions during training, DRIFT samples trajectories offline from a fixed reference policy $\pi_{\mathrm{ref}}$. Crucially, to align with the optimal RL objective derived

in Eq. (9), we concurrently compute a scalar importance weight $w(\tau)$ for each trajectory. This process, detailed in Algorithm 1, efficiently transforms return signals into a high-quality, importance-weighted dataset.

**Stage 2: Weighted Supervised Optimization.** The policy $\pi_\theta$ is then updated via weighted supervised fine-tuning on this pre-collected dataset. DRIFT avoids the extra overhead of repeatedly performing rollouts during the updates of online RL methods. This is particularly critical in multi-turn settings, where the cost of a single rollout grows with the interaction horizon and can be substantially higher than that of a single-turn rollout. Meanwhile, the theoretical guarantees of the KL-regularized objective enable DRIFT to achieve performance close to strong multi-turn RL baselines.

### 4.2. Stage 1: Offline Trajectory Generation

The primary goal of this stage is to construct a high-quality dataset $\mathcal{D}$ that allows us to approximate expectations under the optimal distribution $p^\star$ using samples from the reference policy $\pi_{\text{ref}}$. As detailed in Lines 1–17 of Algorithm 1, this process involves three key components:

**Interaction Protocol.** For each problem prompt $x_1$, we sample $K$ trajectories $\{\tau^{(k)}\}_{k=1}^K$ from the reference policy $\pi_{\text{ref}}$ under a deterministic multi-turn protocol. At any turn $t < T$, if the response $y_t$ is incorrect, a fixed lightweight feedback $f$ is deterministically appended to construct the next state:

$$x_{t+1} = \text{concat}(x_t, y_t, f). \tag{11}$$

The interaction terminates immediately upon generating a correct answer or reaching the maximum turn budget $T$. This protocol ensures that the collected trajectories explicitly capture the model ability to recover from errors under fixed feedback.

**Return Computation.** To strictly align the model behavior with the desired multi-turn outcome, we design a shaped return $R(\tau)$ that incorporates both efficiency and diversity. Let $L$ denote the effective length of trajectory $\tau$. The return is defined as:

$$R(\tau) \triangleq \mathbb{I}(y_L \text{ is correct}) \cdot \gamma^{L-1} - \lambda \left(1 - \frac{E(\tau)}{L}\right), \tag{12}$$

where $\gamma \in (0, 1)$ serves as a discount factor to prioritize solving problems in fewer turns, and the second term imposes a penalty based on the unique response count $E(\tau)$ to discourage repetitive errors, a design inspired by UFO (Liu et al., 2025). We analyze its effect in Section D.8.

**Importance Weight Calculation.** To operationalize the theoretical equivalence established in Theorem 5, we assign a scalar importance weight $w^{(k)}$ to each trajectory. This weight acts as the Radon-Nikodym derivative that adjusts

---

**Algorithm 1** Trajectory Generation in DRIFT

**Require:** Prompts $\{x_1\}$, reference policy $\pi_{\text{ref}}$, parameters $K, \beta$, feedback mechanism
**Ensure:** Weighted dataset $\mathcal{D}$

1: Initialize $\mathcal{D} \leftarrow \emptyset$
2: **for** each prompt $x_1$ **do**
3:      *// Step 1: Sample multiple trajectories*
4:      Sample $K$ trajectories $\{\tau^{(k)}\}_{k=1}^K$ from $\pi_{\text{ref}}$ given $x_1$
5:      *// Step 2: Compute returns and normalization*
6:      **for** $k = 1$ to $K$ **do**
7:          Compute return $R(\tau^{(k)})$ using Eq. (12)
8:      **end for**
9:      Compute $\widehat{Z}(x_1) = \frac{1}{K}\sum_{k=1}^K \exp(R(\tau^{(k)})/\beta)$
10:      *// Step 3: Assign weights and store the final turn*
11:      **for** $k = 1$ to $K$ **do**
12:          Calculate weight $w^{(k)} \leftarrow \frac{\exp(R(\tau^{(k)})/\beta)}{\widehat{Z}(x_1)}$
13:          Let $L^{(k)}$ denote the effective length of $\tau^{(k)}$
14:          Let $(x_{L^{(k)}}, y_{L^{(k)}})$ be the final turn of $\tau^{(k)}$
15:          $\mathcal{D} \leftarrow \mathcal{D} \cup \{(x_{L^{(k)}}, y_{L^{(k)}}, w^{(k)})\}$
16:      **end for**
17: **end for**
18: **return** $\mathcal{D}$

---

the sampling distribution:

$$
\begin{aligned}
w^{(k)} &\leftarrow \frac{\exp\left(R(\tau^{(k)})/\beta\right)}{\widehat{Z}(x_1)}, \\
\widehat{Z}(x_1) &\triangleq \frac{1}{K}\sum_{j=1}^K \exp\left(\frac{R(\tau^{(j)})}{\beta}\right).
\end{aligned} \tag{13}
$$

Here, the partition function estimate $\widehat{Z}(x_1)$ provides prompt-level normalization. This ensures that the weights reflect the relative quality of trajectories within the specific prompt's solution space, stabilizing the variance of the importance sampling estimator.

**Terminal-Step Retention.** Finally, to construct the dataset $\mathcal{D}$, we retain only the terminal turn $(x_L, y_L)$ of each trajectory paired with its weight $w^{(k)}$. This is a protocol-specific approximation to the full-trajectory weighted objective in Theorem 5, rather than an exact implementation of that objective. Under the stop-on-success protocol, intermediate turns in a successful trajectory are verifier-rejected attempts generated under fixed negative feedback. Applying the same large trajectory weight to all such intermediate turns can therefore assign positive credit to responses that did not realize success. Terminal-only retention still conditions on the full interaction history through $x_L$, but concentrates the trajectory weight on the final response that determines the verifier outcome. We analyze and empirically validate this approximation in Section 4.4.

## 4.3. Stage 2: Weighted Supervised Optimization

Once the importance-weighted dataset $\mathcal{D}$ is constructed, the second stage focuses on distilling the optimal behavior into the parameterized policy $\pi_\theta$. This phase effectively converts the reinforcement learning problem into a standard supervised learning task, characterized by the following three aspects:

**Optimization Objective.** Intuitively, we aim to optimize the policy as if it were learning from the theoretical optimal distribution $p^\star$. Since we only have access to trajectories from the reference policy, the importance weights act as a statistical correction mechanism, effectively reshaping the offline data to serve as a proxy for the ideal behavior. Formally, we optimize $\pi_\theta$ by minimizing the weighted negative log-likelihood:

$$\mathcal{L}(\theta) = \mathbb{E}_{(x,y,w)\sim\mathcal{D}}\left[-w \cdot \log \pi_\theta(y \mid x)\right]. \quad (14)$$

This objective is not heuristically chosen; rather, it serves as a Monte Carlo approximation of the cross-entropy between the optimal distribution $p^\star$ and the policy $\pi_\theta$. As derived in Eq. (5), minimizing this weighted loss is theoretically equivalent to minimizing the forward KL divergence $\mathrm{KL}(p^\star\|\pi_\theta)$, thereby ensuring the policy converges towards the optimal solution implied by the KL-regularized RL target.

**Token-Level Realization.** Since the trajectory return is a holistic outcome of the entire generation sequence, we must propagate the trajectory-level evaluation to individual generation steps. In the context of autoregressive models, we implement this by applying the scalar weight $w$ uniformly to the loss of every token in the terminal response $y$:

$$-\log \pi_\theta(y \mid x) = \sum_{j=1}^{M} -\log \pi_\theta(y_j \mid x, y_{<j}), \quad (15)$$

where $M$ is the sequence length. This formulation renders our optimization mechanically equivalent to standard weighted Supervised Fine-Tuning (SFT). This mechanism explicitly amplifies the gradients for desirable reasoning paths while suppressing suboptimal ones, effectively incentivizing the model to assign higher probability mass to tokens that belong to efficient and correct solutions.

A critical advantage of this design is the complete decoupling of rollout generation from parameter updates. Unlike online RL algorithms (e.g., PPO) that require computationally expensive trajectory sampling at every training step, DRIFT treats the weighted dataset $\mathcal{D}$ as a fixed offline corpus. This allows the optimization phase to enjoy the high throughput and stability of standard Supervised Fine-Tuning (SFT), making it significantly more scalable for large language models.

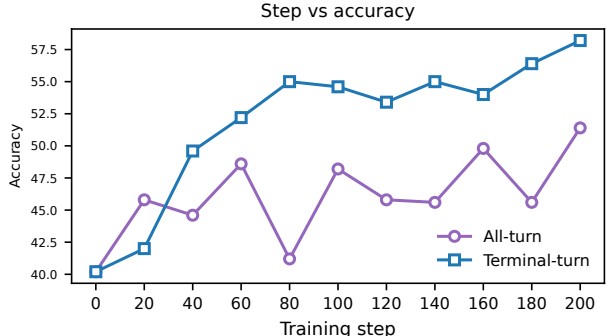

*Figure 3.* Empirical support for terminal-step retention. Both variants use the same offline trajectories and trajectory weights; the all-turn variant supervises every response, while terminal-only supervision uses only the final response conditioned on the full interaction history.

## 4.4. Protocol-Specific Motivation for Terminal-Step Retention

Theorem 5 gives a full-trajectory weighted objective. In the practical implementation of DRIFT, however, we apply the trajectory weight only to the terminal response $(x_L, y_L)$. This terminal-only loss is not an exact implementation of the full-trajectory objective; it is a protocol-specific approximation for the stop-on-success correction setting.

The motivation is credit assignment. Since the episode terminates immediately after the first correct response, every non-terminal response has been rejected by the verifier under the fixed feedback protocol. Applying the same large trajectory weight to all turns may therefore assign positive imitation signal to rejected attempts. Terminal-step retention avoids this issue by supervising only the final response, while still keeping the full interaction history in the conditioning state $x_L$. We formalize this as a bias-variance motivation below.

**Gradient Decomposition.** Let $\ell_t(\theta) \triangleq -\log \pi_\theta(y_t \mid x_t)$. The full-trajectory and terminal-only gradient estimators are

$$g_{\mathrm{all}} \triangleq w(\tau) \sum_{t=1}^{L} \nabla_\theta \ell_t, \qquad g_{\mathrm{term}} \triangleq w(\tau) \nabla_\theta \ell_L. \quad (16)$$

Their difference is

$$\Delta g \triangleq g_{\mathrm{all}} - g_{\mathrm{term}} = w(\tau) \sum_{t=1}^{L-1} \nabla_\theta \ell_t, \quad (17)$$

which corresponds to the omitted gradients from intermediate verifier-rejected responses.

**Proposition 6** (Bias-variance motivation for terminal-step retention). *Assume that $0 \le w(\tau) \le W_{\max}$ and $\|\nabla_\theta \ell_t\| \le G_{\max}$. Then the difference between the full-trajectory and terminal-only expected gradients is bounded by*

$$\|\mathbb{E}[g_{\mathrm{all}}] - \mathbb{E}[g_{\mathrm{term}}]\| \le W_{\max} G_{\max} \mathbb{E}[L - 1]. \quad (18)$$

*Table 1.* Cross-benchmark generalization after training on MetaMathQA (MATH subset). We report **multi@5** accuracy (%) with a maximum budget of 5 turns.

| Model | Method | Math | | | | General | | | | All |
|---|---|---|---|---|---|---|---|---|---|---|
| | | MATH | MATH500 | ThmQA | Avg | MMLU-R | MMLU-P | GPQA | Avg | Avg |
| | | *Single-turn baselines* | | | | | | | | |
| | Base | 38.3 | 40.2 | 26.0 | 34.8 | 76.8 | 49.0 | 47.9 | 57.9 | 46.4 |
| | SFT | $51.2_{\uparrow12.9}$ | $50.8_{\uparrow10.6}$ | $29.8_{\uparrow3.8}$ | $43.9_{\uparrow9.1}$ | $77.0_{\uparrow0.2}$ | $50.4_{\uparrow1.4}$ | $51.0_{\uparrow3.1}$ | $59.5_{\uparrow1.6}$ | $51.7_{\uparrow5.3}$ |
| | PPO | $53.5_{\uparrow15.2}$ | $51.2_{\uparrow11.0}$ | $29.0_{\uparrow3.0}$ | $44.6_{\uparrow9.8}$ | $78.8_{\uparrow2.0}$ | $49.7_{\uparrow0.7}$ | $45.1_{\downarrow2.8}$ | $57.9_{\rightarrow0.0}$ | $51.2_{\uparrow4.8}$ |
| Qwen2.5-3B | | *Multi-turn baselines (Offline)* | | | | | | | | |
| -Instruct | SFT-5turn | $53.3_{\uparrow15.0}$ | $53.6_{\uparrow13.4}$ | $33.1_{\uparrow7.1}$ | $46.7_{\uparrow11.9}$ | $78.2_{\uparrow1.4}$ | $53.3_{\uparrow4.3}$ | $58.0_{\uparrow10.1}$ | $63.2_{\uparrow5.3}$ | $54.9_{\uparrow8.5}$ |
| | STaR-2turn | $50.2_{\uparrow11.9}$ | $50.4_{\uparrow10.2}$ | $29.5_{\uparrow3.5}$ | $43.4_{\uparrow8.6}$ | $84.4_{\uparrow7.6}$ | $57.1_{\uparrow8.1}$ | $58.6_{\uparrow10.7}$ | $66.7_{\uparrow8.8}$ | $55.1_{\uparrow8.7}$ |
| | | *Multi-turn baselines (Online)* | | | | | | | | |
| | SCoRe-2turn | $53.0_{\uparrow14.7}$ | $55.8_{\uparrow15.6}$ | $31.8_{\uparrow5.8}$ | $46.9_{\uparrow12.1}$ | $85.4_{\uparrow8.6}$ | $55.2_{\uparrow6.2}$ | $63.2_{\uparrow15.3}$ | $67.9_{\uparrow10.0}$ | $57.4_{\uparrow11.0}$ |
| | UFO-5turn | $55.5_{\uparrow17.2}$ | $56.4_{\uparrow16.2}$ | $33.5_{\uparrow7.5}$ | $48.5_{\uparrow13.7}$ | $\mathbf{87.0}_{\uparrow10.2}$ | $57.1_{\uparrow8.1}$ | $71.2_{\uparrow23.3}$ | $71.8_{\uparrow13.9}$ | $60.2_{\uparrow13.8}$ |
| | | *Ours* | | | | | | | | |
| | DRIFT-5turn | $\mathbf{55.9}_{\uparrow17.6}$ | $\mathbf{58.2}_{\uparrow18.0}$ | $\mathbf{34.3}_{\uparrow8.3}$ | $\mathbf{49.5}_{\uparrow14.7}$ | $84.6_{\uparrow7.8}$ | $\mathbf{57.2}_{\uparrow8.2}$ | $\mathbf{72.7}_{\uparrow24.8}$ | $\mathbf{71.5}_{\uparrow13.6}$ | $\mathbf{60.5}_{\uparrow14.1}$ |
| | | *Single-turn baselines* | | | | | | | | |
| | Base | 36.8 | 38.2 | 22.0 | 32.3 | 84.5 | 62.0 | 44.9 | 63.8 | 48.1 |
| | SFT | $42.4_{\uparrow5.6}$ | $42.0_{\uparrow3.8}$ | $24.3_{\uparrow2.3}$ | $36.2_{\uparrow3.9}$ | $85.3_{\uparrow0.8}$ | $58.9_{\downarrow3.1}$ | $46.4_{\uparrow1.5}$ | $63.5_{\downarrow0.3}$ | $49.9_{\uparrow1.8}$ |
| | PPO | $40.9_{\uparrow4.1}$ | $41.6_{\uparrow3.4}$ | $30.1_{\uparrow8.1}$ | $37.5_{\uparrow5.2}$ | $87.0_{\uparrow2.5}$ | $60.0_{\downarrow2.0}$ | $50.2_{\uparrow5.3}$ | $65.7_{\uparrow1.9}$ | $51.6_{\uparrow3.5}$ |
| Llama3.1-8B | | *Multi-turn baselines (Offline)* | | | | | | | | |
| -Instruct | SFT-5turn | $41.3_{\uparrow4.5}$ | $44.0_{\uparrow5.8}$ | $23.3_{\uparrow1.3}$ | $36.2_{\uparrow3.9}$ | $85.4_{\uparrow0.9}$ | $57.4_{\downarrow4.6}$ | $48.5_{\uparrow3.6}$ | $63.8_{\rightarrow0.0}$ | $50.0_{\uparrow1.9}$ |
| | STaR-2turn | $41.7_{\uparrow4.9}$ | $43.2_{\uparrow5.0}$ | $26.2_{\uparrow4.2}$ | $37.0_{\uparrow4.7}$ | $86.1_{\uparrow1.6}$ | $62.3_{\uparrow0.3}$ | $55.7_{\uparrow10.8}$ | $68.0_{\uparrow4.2}$ | $52.5_{\uparrow4.4}$ |
| | | *Multi-turn baselines (Online)* | | | | | | | | |
| | SCoRe-2turn | $42.8_{\uparrow6.0}$ | $45.0_{\uparrow6.8}$ | $26.1_{\uparrow4.1}$ | $38.0_{\uparrow5.7}$ | $86.6_{\uparrow2.1}$ | $62.1_{\uparrow0.1}$ | $54.8_{\uparrow9.9}$ | $67.8_{\uparrow4.0}$ | $52.9_{\uparrow4.8}$ |
| | UFO-5turn | $43.1_{\uparrow6.3}$ | $46.4_{\uparrow8.2}$ | $\mathbf{30.3}_{\uparrow8.3}$ | $39.9_{\uparrow7.6}$ | $\mathbf{90.9}_{\uparrow6.4}$ | $\mathbf{64.3}_{\uparrow2.3}$ | $\mathbf{61.6}_{\uparrow16.7}$ | $\mathbf{72.3}_{\uparrow8.5}$ | $\mathbf{56.1}_{\uparrow8.0}$ |
| | | *Ours* | | | | | | | | |
| | DRIFT-5turn | $\mathbf{45.4}_{\uparrow8.6}$ | $\mathbf{48.2}_{\uparrow10.0}$ | $29.9_{\uparrow7.9}$ | $\mathbf{41.2}_{\uparrow8.9}$ | $87.6_{\uparrow3.1}$ | $62.6_{\uparrow0.6}$ | $60.1_{\uparrow15.2}$ | $70.1_{\uparrow6.3}$ | $55.6_{\uparrow7.5}$ |

*Moreover, let* $\mathrm{Var}(z) = \mathbb{E}\|z - \mathbb{E}z\|_2^2$ *and* $\mathrm{Cov}(a,b) = \mathbb{E}\langle a - \mathbb{E}a, b - \mathbb{E}b\rangle$ *for vector-valued gradients. If*

$$\mathrm{Var}(\Delta g) + 2\mathrm{Cov}(g_{\mathrm{term}}, \Delta g) > 0, \quad (19)$$

*then*

$$\mathrm{Var}(g_{\mathrm{term}}) < \mathrm{Var}(g_{\mathrm{all}}). \quad (20)$$

**Remark.** Proposition 6 should be interpreted as a bias-variance motivation rather than an optimizer-equivalence result. Terminal-only retention introduces bias relative to the full-trajectory objective by omitting intermediate turns as imitation targets. Under our stop-on-success protocol, however, those turns are verifier-rejected attempts. When their gradients are noisy or misaligned with the terminal correction signal, terminal-only retention can reduce variance and improve credit assignment. This approximation is therefore justified by the protocol structure.

**Empirical results.** On MATH500 with Qwen2.5-3B-Instruct, we test this interpretation by comparing DRIFT with an all-turn variant that uses the same offline trajectories and trajectory weights, but applies each weight to every response in the trajectory. As shown in Figure 3, terminal-only

supervision reaches higher accuracy and yields a smoother optimization curve under the same training schedule. This supports the prediction of Proposition 6: in stop-on-success trajectories, retroactively assigning large weights to verifier-rejected intermediate responses can amplify misaligned gradients, whereas concentrating supervision on the terminal response provides a better bias-variance tradeoff in practice.

## 5. Experiments

### 5.1. Experimental Setup

**Training setup.** We employ Qwen2.5-3B-Instruct (Qwen et al., 2025) and Llama3.1-8B-Instruct (Grattafiori et al., 2024) as the base model and train it on the MATH subset of the MetaMathQA (Yu et al., 2024) dataset. We introduce more training details in Section D.2.

**Benchmark.** All evaluations are performed using greedy decoding. Since all methods are trained on the MATH subset of MetaMathQA, we organize the evaluation benchmarks into two groups: math reasoning benchmarks from the same broad task family, and out-of-domain general reasoning benchmarks. This split tests whether learned multi-turn

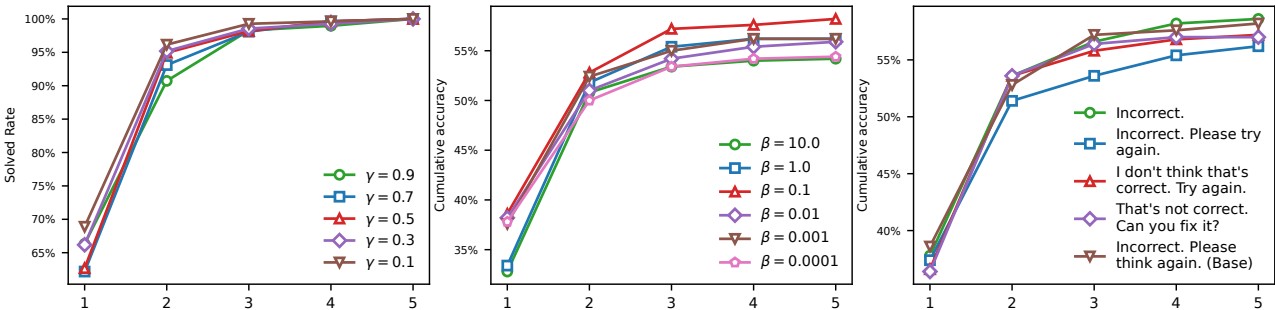

*Figure 4.* For different $\gamma$ values, report the proportion of problems cumulatively solved at each turn relative to the total number of problems solved by the end. For different $\beta$ and feedback values, report the cumulative accuracy at each turn.

correction transfers beyond the training domain rather than only improving MATH-style problem solving. We detail the benchmarks and experimental settings in Section D.3.

**Metrics.** We report the cumulative accuracy with a maximum budget of 5 turns, denoted as multi@5. Generation will terminate immediately if the answer is correct, otherwise it continues up to $k$ turns. This metric is formally defined as:

$$\text{multi@}k \;=\; \frac{1}{N} \sum_{i=1}^{N} \max_{t \in \{1,\ldots,k\}} \mathcal{V}(y_{i,t}). \qquad (21)$$

where $\mathcal{V}(\cdot)$ denotes the verifier function and $N$ is the number of test samples.

### 5.2. Main Results

Table 1 summarizes the main results. Across most benchmarks, the RL-based method UFO outperforms SFT-based baselines, and DRIFT further improves upon UFO under matched settings. All methods are trained on math data. Single-turn training attains multi@k performance close to multi-turn training on MATH-style benchmarks mainly by improving the first-turn accuracy; however, it does not learn to condition on negative feedback and thus yields little gain on non-math benchmarks. In contrast, multi-turn training explicitly learns correction under negative feedback, leading to substantial improvements and enabling effective multi-turn behavior beyond math tasks. Among multi-turn methods, RL-based approaches remain stronger than SFT-based ones, with the gap most pronounced on non-math benchmarks. By realizing the RL objective with an SFT-style optimization, DRIFT matches or surpasses RL baselines on most benchmarks. In small discrete action spaces (e.g., multiple-choice), UFO significantly suppresses repetition, enabling models to guess via elimination when intrinsic capability is lacking, notably on Llama-3.1-8B-Instruct. We illustrate examples of blind guessing and briefly discuss them in Section E.3.

### 5.3. Training efficiency comparison

We compare training efficiency for Qwen2.5-3B-Instruct and Llama3.1-8B-Instruct on two hardware configurations: $4\times$ NVIDIA A800 (80GB) GPUs and $4\times$ NVIDIA H20 (96GB) GPUs, and report GPU time. GPU time measures the end-to-end wall-clock time, including rollout latency for methods that perform rollouts (both UFO and DRIFT). We run 200 training steps with a global batch size of 128 distributed over 4 GPUs. As the number of turns increases, SFT-5Turn and DRIFT incur only a small increase in GPU time, while UFO-5Turn becomes substantially slower due to the growing cost of multi-turn rollouts. Overall, DRIFT achieves higher training-time efficiency than UFO, and its scaling behavior remains close to SFT-style training as the turn count grows.

### 5.4. Turn-by-turn performance comparison

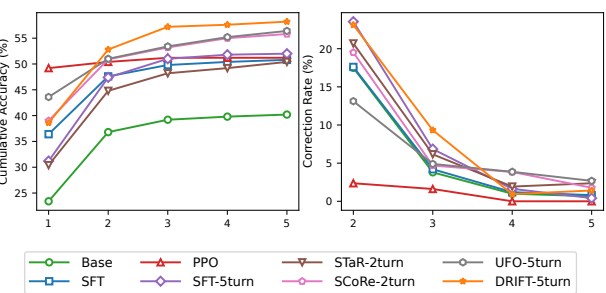

*Figure 5.* Cumulative success rate and correction rate per turn on MATH500 for Qwen2.5-3B-Instruct trained with different methods.

In Fig. 5, we report the cumulative accuracy and correction rate (# correct this turn / # wrong in the previous turn) at each turn after training for different methods. Among all methods, DRIFT achieves a significantly higher correction rate in early turns. And the results show that single-turn training primarily improves the turn 1 accuracy, with only marginal gains in later turns. In particular, the model trained with single-turn PPO is almost unable to continue correcting its answers under multi-turn interactions. Among multi-turn

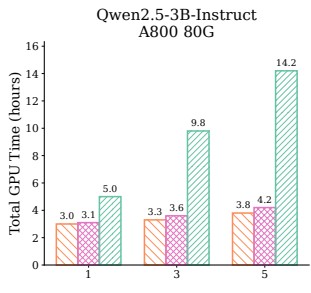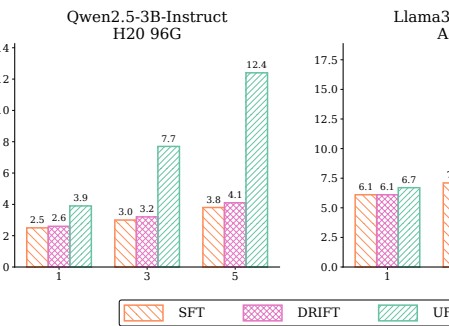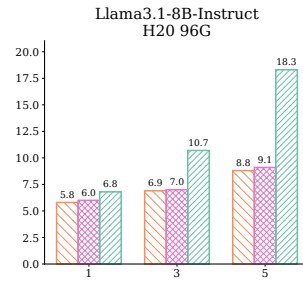

*Figure 6.* Training efficiency comparison in GPU time across two base models and two hardware configurations. Compare with multi-turn SFT (SFT-5Turn) and multi-turn RL (UFO-5Turn).

training methods, RL-based approaches exhibit stronger turn-by-turn performance than SFT-based ones. However, as shown in Fig. 6, RL-based methods are substantially less training-efficient. In contrast, DRIFT achieves multi-turn improvements that match or surpass RL-based methods, while retaining a training efficiency comparable to SFT-based methods.

### 5.5. Impact of hyperparameters

We analyze the effect of different hyperparameters $\gamma$ and $\beta$, as shown in Fig. 4. To compare convergence behaviors across different $\gamma$ values, we define the *solved rate* as the ratio of the cumulative success rate at a given turn to the final success rate achieved by the end of the episode. This metric indicates the fraction of ultimately solvable problems that are solved by the current turn. The results demonstrate that a smaller $\gamma$ encourages the model to solve problems in fewer turns, aligning with the discount factor's role in incentivizing early success.

For different $\beta$ values, we report the cumulative accuracy at each turn in Fig. 4 (middle). The results show that a moderate $\beta$ (we use $\beta = 0.1$) performs best: larger $\beta$ leads to weaker exponential tilting and smaller multi-turn gains, while smaller $\beta$ makes the weights overly concentrated and degrades stability. This matches the role of $\beta$ in Theorem 1 (controlling the sharpness of the tilted distribution) and the stability trade-off discussed in Proposition 7.

Although we train with negative feedback string $f =$ "Incorrect. Please think again.", the learned correction behavior is not sensitive to the exact wording. As shown in Fig. 4 (right), different feedback strings $f$ have only a minor impact on performance and simpler feedback such as "Incorrect" can better elicit the model's multi-turn capability, suggesting stable performance across different feedback variants.

## 6. Conclusion & Limitations

**Conclusion.** In this work, we introduced DRIFT, a practi-

cal framework for multi-turn optimization under lightweight negative feedback. DRIFT decouples trajectory generation from optimization by sampling multi-turn correction trajectories once from a frozen reference policy, assigning each trajectory an exponential weight derived from a KL-regularized multi-turn objective, and training the target model using an importance-weighted supervised fine-tuning objective. This formulation provides a simple and stable alternative to online multi-turn reinforcement learning, avoiding repeated rollouts during training while still optimizing for both accuracy and efficiency across turns. Empirically, under matched settings, DRIFT achieves performance comparable to or better than strong multi-turn RL baselines across math and general-domain benchmarks, while approaching the efficiency profile of standard supervised fine-tuning as the interaction horizon increases.

**Limitations.** DRIFT is intended for short-horizon, verifier-guided correction with lightweight deterministic feedback. This boundary matches our problem formulation and evaluation protocol: (1) The verifier provides an unambiguous correctness signal, and each episode contains only a small number of correction attempts. Settings with stochastic or preference-based human feedback, open-ended dialogue objectives, or substantially longer-horizon interactive planning require additional modeling of feedback uncertainty, credit assignment, and exploration, and are left for future work. (2) A second limitation comes from offline rollout coverage. Because DRIFT samples correction trajectories from a fixed reference policy and does not alternate rollout collection with policy optimization, it can only upweight useful behaviors that appear in the collected trajectories. It may therefore miss strategies that online RL could discover through repeated exploration. As a simple initial attempt to reduce this limitation, Appendix D.5 studies a two-stage rollout-refresh variant that regenerates rollouts after an intermediate DRIFT checkpoint and obtains a modest gain over single-stage DRIFT. More systematic rollout-refresh schedules and hybrid offline-online training remain open directions.

## Impact Statement

This paper presents work whose goal is to advance the field of Machine Learning. There are many potential societal consequences of our work, none which we feel must be specifically highlighted here.

## Acknowledgements

This work was supported in part by the National Natural Science Foundation of China (Grant Nos. 62506319), the Guangdong Basic and Applied Basic Research Foundation (Grant No. 2026A1515030032), the Shenzhen Science and Technology Program (Grant No. JCYJ20250604141031003), and the Pearl River Talent Program of Guangdong Province (Grant No. 2024QN11X069).

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

# A. Related Work

**Multi-turn self-correction under lightweight negative feedback.** Large language models (LLMs) exhibit a non-trivial but fragile ability to revise their answers when presented with minimal negative feedback such as "Incorrect, please try again" (Liu et al., 2025; Kumar et al., 2024). However, this behavior is unreliable. Extended multi-turn contexts can cause models to drift or "get lost" as dialogue history grows (Laban et al., 2025; Li et al., 2025), and intrinsic self-correction may degrade answer quality, produce superficial edits, or amplify errors (Zhang et al., 2025; Li et al., 2024). Beyond prompting-based elicitation, a line of work studies iterative improvement via self-critique and refinement, for example self-refinement and reflection-style agents (Madaan et al., 2023; Shinn et al., 2023). These approaches can help, but they typically do not optimize a principled trajectory-level objective and they remain sensitive to prompting, verification, and context accumulation. Empirically, multi-step reasoning can sometimes be elicited by generic "try again" messages (Liu et al., 2025), but the gains vary by task and model family, motivating learning objectives that directly target multi-turn success rather than only first-turn correctness. These observations align with our setting, a deterministic multi-turn correction protocol with lightweight negative feedback, and motivate optimizing the trajectory-level outcome that DRIFT is designed to improve.

**Learning from correction trajectories and multi turn fine tuning.** A common approach to improve revision behavior is to collect correction traces and apply supervised fine-tuning (SFT) on multi-turn demonstrations (Zelikman et al., 2022; Welleck et al., 2022; Singh et al., 2023). Self-training and bootstrapping methods such as STaR generate candidate solutions or rationales and then distill them into the model via SFT, improving problem solving with limited additional supervision (Zelikman et al., 2022; Singh et al., 2023). For interactive correction, offline multi-turn SFT can teach models to condition on feedback, but it can also suffer from distribution shift. The model trains on traces whose early turns are produced by a different policy than the one deployed at test time, so correction quality may drop when the first-turn response is generated by the updated model (Kumar et al., 2024). Moreover, SFT can exhibit behavioral collapse in which the model over-optimizes first-attempt accuracy while performing shallow or non-committal revisions in later turns (Kumar et al., 2024). Recent analyses further suggest that intermediate turns may contain noisy or spurious reasoning paths, complicating stable learning from full trajectories (Zhang et al., 2025). Complementary to multi-turn trace SFT, recent offline RL formulations for conversational policies recast short-horizon interaction learning as return-weighted fine-tuning, enabling direct use of scalar rewards on logged dialogue trajectories (Mukherjee et al., 2025). In contrast to unweighted multi-turn SFT on fixed traces, DRIFT retains SFT-like optimization but reweights offline trajectories to approximate the intended multi-turn objective, thereby addressing these failure modes while staying compatible with efficient fine-tuning pipelines.

**KL regularized policy optimization and return weighted fine tuning.** To optimize multi-turn outcomes, reinforcement learning (RL) and RLHF update the policy from rollouts and reward signals, often with KL regularization to stay close to a reference model (Schulman et al., 2017; Zheng et al., 2025a; Ouyang et al., 2022; Gao et al., 2024). In LLM alignment, PPO-style RLHF and sequence-level variants such as GRPO (Shao et al., 2024; Zheng et al., 2025b) can be viewed as KL-regularized distribution matching, while single-turn alternatives such as DPO replace online RL with closed-form preference objectives (Rafailov et al., 2023). More broadly, RL-as-inference and reward-weighted or advantage-weighted regression show that KL-regularized objectives induce exponential reweighting of actions or trajectories (Peters & Schaal, 2007; Levine, 2018; Peng et al., 2019; Nair et al., 2020), enabling offline optimization via importance weighting. Related connections also appear in structured prediction, where reward augmented maximum likelihood (RAML) integrates task rewards into maximum likelihood training through an exponentiated reward distribution (Norouzi et al., 2016). Building on these views, recent alignment methods derive reward-driven reweighted SFT objectives through variational or bound-tightening analyses, including VAR (Du et al., 2026) and importance-weighted SFT (Qin & Springenberg, 2025), and successive policy reweighting schemes that target RL objectives with SFT-like compute (Zhang et al., 2024). Complementary analyses study when RLVR with answer-only rewards can still promote correct reasoning and what normalization or supervision improves stability (Wen et al., 2025). Although online multi-turn RL can be effective (Gao et al., 2024; Kumar et al., 2024), its rollout cost grows quickly with horizon because each update requires full multi-turn trajectories. DRIFT applies these KL-regularized, weighted-likelihood principles at the trajectory level using offline rollouts under a reference policy, prompt-level weight normalization, and terminal-only supervision, decoupling rollout from optimization and achieving an RL-motivated target with SFT-like efficiency.

## B. Additional theoretical analyses

### B.1. Estimation Stability and Sample Complexity

The stability of the gradient estimator relies on the accuracy of the normalized weights $\widehat{w}(\tau) = \exp(R(\tau)/\beta)/\widehat{Z}(x)$. Since the Monte Carlo estimate $\widehat{Z}(x)$ appears in the denominator, small estimation errors can be amplified, particularly when the partition function is small. Here, we analyze the sample complexity required to bound this error.

**Proposition 7** (Concentration of the Partition Estimate). *Assume bounded returns $R(\tau) \in [R_{\min}, R_{\max}]$. Let $m_\beta \triangleq \exp(R_{\min}/\beta)$ and $M_\beta \triangleq \exp(R_{\max}/\beta)$ be the bounds of the exponentiated return, with range $\Delta_\beta \triangleq M_\beta - m_\beta$. For any $\epsilon > 0$, Hoeffding's inequality guarantees:*

$$P\left(\left|\widehat{Z}(x) - Z(x)\right| \geq \epsilon\right) \leq 2\exp\left(-\frac{2K\epsilon^2}{\Delta_\beta^2}\right). \tag{22}$$

*To ensure $|\widehat{Z}(x) - Z(x)| \leq \epsilon$ with probability at least $1 - \delta$, the sample size $K$ must satisfy:*

$$K \geq \frac{\Delta_\beta^2}{2\epsilon^2} \log\left(\frac{2}{\delta}\right). \tag{23}$$

**Stability of Normalized Weights.** Bounding the error of $\widehat{Z}$ is a prerequisite for stable training. Since $Z(x) \geq m_\beta > 0$, the function $f(z) = 1/z$ is Lipschitz continuous on $[m_\beta, \infty)$ with constant $1/m_\beta^2$. Consequently, an estimation error $|\widehat{Z} - Z| \leq \epsilon$ propagates to the importance weights as:

$$|\widehat{w}(\tau) - w(\tau)| \leq M_\beta \left|\frac{1}{\widehat{Z}} - \frac{1}{Z}\right| \leq \frac{M_\beta}{m_\beta^2}\epsilon. \tag{24}$$

This confirms that sufficiently large $K$ ensures the convergence of the empirical weights to their theoretical values.

**The Regularization-Complexity Trade-off.** Eq. (23) reveals that the sample complexity is governed by the range $\Delta_\beta$. In the *weak regularization* regime ($\beta \to 0$), $\Delta_\beta$ grows exponentially, indicating that in the worst case, the required $K$ to maintain a fixed precision $\epsilon$ scales with $\exp(2R_{\max}/\beta)$. This theoretical bound justifies our use of a moderate $\beta$: it balances the sharpness of the distribution matching against the need for sample efficiency, preventing the gradient estimator from being dominated by high-variance sampling noise.

## C. Proofs.

### C.1. Proof of Thm 1

*Proof.* Fix an initial prompt $x$ and abbreviate $p(\tau) \triangleq p(\tau \mid x)$ and $p_{\text{ref}}(\tau) \triangleq p_{\text{ref}}(\tau \mid x)$. Let $\mathcal{T}_x$ denote the (countable) set of valid trajectories under the deterministic protocol. Consider the variational problem

$$\max_{p(\cdot)} \quad \sum_{\tau \in \mathcal{T}_x} p(\tau)\, R(\tau) \; - \; \beta \sum_{\tau \in \mathcal{T}_x} p(\tau) \log \frac{p(\tau)}{p_{\text{ref}}(\tau)} \tag{25}$$

$$\text{s.t.} \quad \sum_{\tau \in \mathcal{T}_x} p(\tau) = 1, \qquad p(\tau) \geq 0 \;\; \forall\tau.$$

If $p_{\text{ref}}(\tau) = 0$ for some $\tau$, any feasible $p$ with $p(\tau) > 0$ yields $\text{KL}(p\|p_{\text{ref}}) = +\infty$; hence at the optimum we must have $p(\tau) = 0$ whenever $p_{\text{ref}}(\tau) = 0$. In the remainder, we restrict to $\tau$ with $p_{\text{ref}}(\tau) > 0$.

Form the Lagrangian (we temporarily ignore inequality constraints and verify feasibility afterward):

$$\mathcal{L}(p, \lambda) = \sum_\tau p(\tau)R(\tau) - \beta \sum_\tau p(\tau) \log \frac{p(\tau)}{p_{\text{ref}}(\tau)} + \lambda\left(\sum_\tau p(\tau) - 1\right). \tag{26}$$

Taking the derivative w.r.t. $p(\tau)$ and setting it to zero gives, for each $\tau$ with $p_{\text{ref}}(\tau) > 0$,

$$0 = \frac{\partial \mathcal{L}}{\partial p(\tau)} = R(\tau) - \beta\left(\log \frac{p(\tau)}{p_{\text{ref}}(\tau)} + 1\right) + \lambda. \tag{27}$$

Rearranging (27) yields

$$\log \frac{p(\tau)}{p_{\text{ref}}(\tau)} = \frac{R(\tau)}{\beta} + \frac{\lambda - \beta}{\beta}, \qquad \Longrightarrow \qquad p(\tau) = C \, p_{\text{ref}}(\tau) \exp\left(\frac{R(\tau)}{\beta}\right), \tag{28}$$

where $C \triangleq \exp\left((\lambda - \beta)/\beta\right)$ is a constant independent of $\tau$. Imposing the normalization constraint $\sum_{\tau} p(\tau) = 1$ gives

$$1 = \sum_{\tau} p(\tau) = C \sum_{\tau} p_{\text{ref}}(\tau) \exp\left(\frac{R(\tau)}{\beta}\right) = C \, Z(x), \tag{29}$$

so $C = 1/Z(x)$, with

$$Z(x) \triangleq \sum_{\tau \in \mathcal{T}_x} p_{\text{ref}}(\tau \mid x) \exp\left(\frac{R(\tau)}{\beta}\right) = \mathbb{E}_{\tau \sim p_{\text{ref}}(\cdot \mid x)}\left[\exp\left(\frac{R(\tau)}{\beta}\right)\right]. \tag{30}$$

Therefore, the (candidate) maximizer is

$$p^\star(\tau \mid x) = \frac{1}{Z(x)} \, p_{\text{ref}}(\tau \mid x) \exp\left(\frac{R(\tau)}{\beta}\right). \tag{31}$$

Since $R(\tau)$ is bounded by assumption, $\exp(R(\tau)/\beta)$ is bounded, hence $0 < Z(x) < \infty$ and $p^\star$ is well-defined.

It remains to show optimality and uniqueness. The feasible set $\{p : \sum_{\tau} p(\tau) = 1, \, p(\tau) \geq 0\}$ is convex. The mapping $p \mapsto \sum_{\tau} p(\tau) R(\tau)$ is linear, and $p \mapsto -\beta \, \text{KL}(p \| p_{\text{ref}})$ is strictly concave on $\{p : p \ll p_{\text{ref}}\}$. Hence the objective in (25) is strictly concave over the feasible region (restricted to the support of $p_{\text{ref}}$), so any stationary point is the unique global maximizer. Consequently, $p^\star(\cdot \mid x)$ in (31) is the unique solution to (25). $\qquad \square$

## C.2. Proof of Thm 2

*Proof.* Fix a prompt $x$. For brevity, write

$$P_\theta \triangleq p_\theta(\cdot \mid x), \quad P_{\text{ref}} \triangleq p_{\text{ref}}(\cdot \mid x), \quad P^\star \triangleq p^\star(\cdot \mid x).$$

If $P_\theta \not\ll P_{\text{ref}}$, then $\text{KL}(P_\theta \| P_{\text{ref}}) = +\infty$ and the claimed identity holds trivially (both sides equal $-\infty$ under the convention $J(\theta) = -\infty$). Hence assume $P_\theta \ll P_{\text{ref}}$.

Recall that $P^\star$ is defined by

$$p^\star(\tau \mid x) = \frac{1}{Z(x)} \, p_{\text{ref}}(\tau \mid x) \exp\left(\frac{R(\tau)}{\beta}\right), \qquad Z(x) \triangleq \sum_{\tau} p_{\text{ref}}(\tau \mid x) \exp\left(\frac{R(\tau)}{\beta}\right), \tag{32}$$

so $P^\star \ll P_{\text{ref}}$ and

$$\log \frac{p^\star(\tau \mid x)}{p_{\text{ref}}(\tau \mid x)} = \frac{R(\tau)}{\beta} - \log Z(x).$$

Therefore, for any $\tau$ with $p_\theta(\tau \mid x) > 0$,

$$\begin{aligned}
\log \frac{p_\theta(\tau \mid x)}{p^\star(\tau \mid x)} &= \log \frac{p_\theta(\tau \mid x)}{p_{\text{ref}}(\tau \mid x)} - \log \frac{p^\star(\tau \mid x)}{p_{\text{ref}}(\tau \mid x)} \\
&= \log \frac{p_\theta(\tau \mid x)}{p_{\text{ref}}(\tau \mid x)} - \frac{R(\tau)}{\beta} + \log Z(x).
\end{aligned} \tag{33}$$

Taking expectation under $\tau \sim P_\theta$ yields

$$\begin{aligned}
\text{KL}(P_\theta \| P^\star) &\triangleq \mathbb{E}_{\tau \sim P_\theta}\left[\log \frac{p_\theta(\tau \mid x)}{p^\star(\tau \mid x)}\right] \\
&= \mathbb{E}_{\tau \sim P_\theta}\left[\log \frac{p_\theta(\tau \mid x)}{p_{\text{ref}}(\tau \mid x)}\right] - \frac{1}{\beta} \mathbb{E}_{\tau \sim P_\theta}\left[R(\tau)\right] + \log Z(x) \\
&= \text{KL}(P_\theta \| P_{\text{ref}}) - \frac{1}{\beta} \mathbb{E}_{\tau \sim P_\theta}\left[R(\tau)\right] + \log Z(x).
\end{aligned} \tag{34}$$

Rearranging (34) gives

$$\mathbb{E}_{\tau \sim P_\theta}\big[R(\tau)\big] - \beta \, \mathrm{KL}(P_\theta \| P_{\mathrm{ref}}) = \beta \log Z(x) - \beta \, \mathrm{KL}(P_\theta \| P^\star), \tag{35}$$

which is exactly the statement of Thm. 2. □

### C.3. Proof of Lemma 3

*Proof.* Fix a prompt $x$ and let $\mathcal{T}_x$ denote the (countable) set of valid trajectories under $x$. For notational simplicity, define

$$P_\theta(\tau) \triangleq p_\theta(\tau \mid x), \qquad P^\star(\tau) \triangleq p^\star(\tau \mid x), \qquad \forall \tau \in \mathcal{T}_x. \tag{36}$$

By the realizability assumption in Lemma 3, there exists $\theta^\star$ such that

$$P_{\theta^\star}(\tau) = P^\star(\tau), \qquad \forall \tau \in \mathcal{T}_x. \tag{37}$$

We will show that both objectives $\mathrm{KL}(P_\theta \| P^\star)$ and $\mathrm{KL}(P^\star \| P_\theta)$ attain their global minimum at exactly the set of parameters satisfying $P_\theta = P^\star$, and moreover minimizing $\mathrm{KL}(P^\star \| P_\theta)$ is equivalent to minimizing the cross-entropy term $\mathbb{E}_{\tau \sim P^\star}\big[ -\log p_\theta(\tau \mid x) \big]$.

**Step 1: Minimizers of the reverse KL.** Recall the definition of KL divergence on $\mathcal{T}_x$:

$$\mathrm{KL}(P \| Q) \triangleq \sum_{\tau \in \mathcal{T}_x} P(\tau) \log \frac{P(\tau)}{Q(\tau)}, \tag{38}$$

with the standard convention that if $P(\tau) > 0$ and $Q(\tau) = 0$ for some $\tau$, then $\mathrm{KL}(P \| Q) = +\infty$.

Therefore, if there exists $\tau \in \mathcal{T}_x$ such that $P_\theta(\tau) > 0$ but $P^\star(\tau) = 0$, then

$$\mathrm{KL}(P_\theta \| P^\star) = +\infty. \tag{39}$$

On the other hand, by (37),

$$\mathrm{KL}(P_{\theta^\star} \| P^\star) = \mathrm{KL}(P^\star \| P^\star) = 0. \tag{40}$$

Hence any $\theta$ satisfying (39) cannot be a minimizer of $\theta \mapsto \mathrm{KL}(P_\theta \| P^\star)$.

Next, for all $\theta$ such that $\mathrm{KL}(P_\theta \| P^\star)$ is finite, KL non-negativity implies

$$\mathrm{KL}(P_\theta \| P^\star) \geq 0. \tag{41}$$

Combining (40) and (41), the global minimum value of $\theta \mapsto \mathrm{KL}(P_\theta \| P^\star)$ is 0.

Moreover, if $\hat{\theta}$ is any minimizer, then

$$\mathrm{KL}(P_{\hat{\theta}} \| P^\star) = 0. \tag{42}$$

By the equality condition of Gibbs' inequality (equivalently, $\mathrm{KL}(P \| Q) = 0$ iff $P = Q$ as distributions on $\mathcal{T}_x$), (42) implies

$$P_{\hat{\theta}}(\tau) = P^\star(\tau), \qquad \forall \tau \in \mathcal{T}_x. \tag{43}$$

Conversely, any $\theta$ satisfying $P_\theta = P^\star$ yields $\mathrm{KL}(P_\theta \| P^\star) = 0$ and thus is a global minimizer. Therefore,

$$\arg\min_\theta \, \mathrm{KL}(P_\theta \| P^\star) = \{\theta : \, P_\theta = P^\star\}. \tag{44}$$

**Step 2: Minimizers of the forward KL.** By the definition (38), if there exists $\tau \in \mathcal{T}_x$ such that $P^\star(\tau) > 0$ but $P_\theta(\tau) = 0$, then

$$\mathrm{KL}(P^\star \| P_\theta) = +\infty. \tag{45}$$

In contrast, by (37),

$$\mathrm{KL}(P^\star \| P_{\theta^\star}) = \mathrm{KL}(P^\star \| P^\star) = 0. \tag{46}$$

Thus any $\theta$ satisfying (45) cannot be a minimizer.

For all $\theta$ such that $\mathrm{KL}(P^\star\|P_\theta)$ is finite, KL non-negativity gives

$$\mathrm{KL}(P^\star\|P_\theta) \geq 0. \tag{47}$$

Combining (46) and (47), the global minimum value is again 0. Hence any minimizer $\tilde{\theta}$ must satisfy

$$\mathrm{KL}(P^\star\|P_{\tilde{\theta}}) = 0, \tag{48}$$

which implies (by the equality condition of Gibbs' inequality)

$$P_{\tilde{\theta}}(\tau) = P^\star(\tau), \qquad \forall \tau \in \mathcal{T}_x. \tag{49}$$

Therefore,

$$\arg\min_{\theta} \ \mathrm{KL}(P^\star\|P_\theta) = \{\theta : \ P_\theta = P^\star\}. \tag{50}$$

**Step 3: Forward-KL minimization is equivalent to cross-entropy minimization.** Fix the same prompt $x$ and consider the objective $\theta \mapsto \mathrm{KL}(P^\star\|P_\theta)$. If there exists $\tau \in \mathcal{T}_x$ such that $P^\star(\tau) > 0$ but $P_\theta(\tau) = 0$, then by definition $\mathrm{KL}(P^\star\|P_\theta) = +\infty$, and also $\mathbb{E}_{\tau \sim P^\star}\big[-\log p_\theta(\tau \mid x)\big] = +\infty$. Hence the claimed equivalence holds trivially. In the remainder assume $P^\star \ll P_\theta$.

By definition of forward KL divergence,

$$\begin{aligned}
\mathrm{KL}\big(P^\star\|P_\theta\big) &\triangleq \sum_{\tau \in \mathcal{T}_x} P^\star(\tau) \log \frac{P^\star(\tau)}{P_\theta(\tau)} \\
&= \sum_{\tau \in \mathcal{T}_x} P^\star(\tau) \log P^\star(\tau) - \sum_{\tau \in \mathcal{T}_x} P^\star(\tau) \log P_\theta(\tau) \\
&= \mathbb{E}_{\tau \sim P^\star}\big[\log p^\star(\tau \mid x)\big] + \mathbb{E}_{\tau \sim P^\star}\big[-\log p_\theta(\tau \mid x)\big].
\end{aligned} \tag{51}$$

The first term in (51) depends only on $P^\star$ (hence is independent of $\theta$). Therefore,

$$\arg\min_{\theta} \ \mathrm{KL}\big(P^\star\|P_\theta\big) \ = \ \arg\min_{\theta} \ \mathbb{E}_{\tau \sim P^\star}\big[-\log p_\theta(\tau \mid x)\big], \tag{52}$$

which proves ( 8). $\qquad\square$

### C.4. Proof of Lemma 4

*Proof.* Fix a prompt $x$. For brevity, write

$$P(\tau) = P^\star(\tau) = p^\star(\tau \mid x), \qquad Q(\tau) = P_\theta(\tau) = p_\theta(\tau \mid x),$$

and let $\mathcal{T}_x = \mathrm{supp}(P)$. Since $\mathcal{T}_x$ is finite and $P$ is positive on its support,

$$m_x := \min_{\tau \in \mathcal{T}_x} P(\tau) > 0.$$

Choose $\varepsilon_x > 0$ small enough such that

$$\mathrm{TV}(Q, P) \leq \varepsilon_x \quad\Longrightarrow\quad \left\|\frac{Q - P}{P}\right\|_\infty \leq \rho$$

for some fixed $\rho < 1$. For example, one may take $\varepsilon_x < m_x/2$. Define the relative perturbation

$$r(\tau) := \frac{Q(\tau) - P(\tau)}{P(\tau)}.$$

Then $Q(\tau) = P(\tau)(1 + r(\tau))$, $\|r\|_\infty \le \rho < 1$, and

$$\sum_{\tau \in \mathcal{T}_x} P(\tau)r(\tau) = \sum_{\tau \in \mathcal{T}_x} (Q(\tau) - P(\tau)) = 0.$$

We now compare the two KL directions. For $|u| \le \rho < 1$, Taylor expansion around $u = 0$ gives, uniformly in $u$,

$$(1 + u)\log(1 + u) = u + \frac{1}{2}u^2 + O_\rho(u^3),$$

and

$$-\log(1 + u) = -u + \frac{1}{2}u^2 + O_\rho(u^3).$$

Therefore,

$$\mathrm{KL}(Q\|P) = \sum_{\tau \in \mathcal{T}_x} P(\tau)(1 + r(\tau))\log(1 + r(\tau))$$

$$= \sum_{\tau \in \mathcal{T}_x} P(\tau)\left[r(\tau) + \frac{1}{2}r(\tau)^2 + O_\rho(|r(\tau)|^3)\right],$$

and

$$\mathrm{KL}(P\|Q) = \sum_{\tau \in \mathcal{T}_x} P(\tau)\left[-\log(1 + r(\tau))\right]$$

$$= \sum_{\tau \in \mathcal{T}_x} P(\tau)\left[-r(\tau) + \frac{1}{2}r(\tau)^2 + O_\rho(|r(\tau)|^3)\right].$$

The first-order terms vanish in both expressions because $\sum_\tau P(\tau)r(\tau) = 0$. Hence both KL divergences share the same quadratic term:

$$\frac{1}{2}\sum_{\tau \in \mathcal{T}_x} P(\tau)r(\tau)^2 = \frac{1}{2}\sum_{\tau \in \mathcal{T}_x} \frac{(Q(\tau) - P(\tau))^2}{P(\tau)}.$$

Their difference is therefore controlled by the third-order remainders:

$$|\mathrm{KL}(Q\|P) - \mathrm{KL}(P\|Q)| \le C_\rho \sum_{\tau \in \mathcal{T}_x} P(\tau)|r(\tau)|^3$$

for some constant $C_\rho < \infty$. Substituting the definition of $r$,

$$\sum_{\tau \in \mathcal{T}_x} P(\tau)|r(\tau)|^3 = \sum_{\tau \in \mathcal{T}_x} \frac{|Q(\tau) - P(\tau)|^3}{P(\tau)^2} \le \frac{1}{m_x^2}\sum_{\tau \in \mathcal{T}_x} |Q(\tau) - P(\tau)|^3.$$

Finally,

$$\sum_{\tau \in \mathcal{T}_x} |Q(\tau) - P(\tau)|^3 \le \left(\sum_{\tau \in \mathcal{T}_x} |Q(\tau) - P(\tau)|\right)^3 = 8\,\mathrm{TV}(Q, P)^3.$$

Combining the above inequalities, there exists a finite constant

$$C_x < \infty$$

depending only on $x$, $P^\star$, and the chosen local neighborhood, such that

$$|\mathrm{KL}(Q\|P) - \mathrm{KL}(P\|Q)| \le C_x\,\mathrm{TV}(Q, P)^3.$$

Substituting back $Q = P_\theta$ and $P = P^\star$ proves the claim.

The shared quadratic term also shows that Forward-KL and Reverse-KL have the same local second-order geometry around $p^\star$, even when $p^\star \notin \Pi_\theta$. $\qquad\square$

## C.5. Proof of Thm 5

*Proof.* Fix a prompt $x$ and let $\mathcal{T}_x$ be the (countable) set of valid trajectories. Let $p^\star(\cdot \mid x)$ be defined as in Thm. 1, i.e.,

$$p^\star(\tau \mid x) = \frac{1}{Z(x)}\, p_{\mathrm{ref}}(\tau \mid x)\exp\!\Big(\frac{R(\tau)}{\beta}\Big), \qquad Z(x) \triangleq \sum_{\tau \in \mathcal{T}_x} p_{\mathrm{ref}}(\tau \mid x)\exp\!\Big(\frac{R(\tau)}{\beta}\Big). \tag{53}$$

In particular, $p^\star(\cdot \mid x) \ll p_{\mathrm{ref}}(\cdot \mid x)$, and the Radon–Nikodym derivative is

$$w(\tau \mid x) \triangleq \frac{p^\star(\tau \mid x)}{p_{\mathrm{ref}}(\tau \mid x)} = \frac{1}{Z(x)}\exp\!\Big(\frac{R(\tau)}{\beta}\Big), \tag{54}$$

for all $\tau$ with $p_{\mathrm{ref}}(\tau \mid x) > 0$ (and we may set $w(\tau \mid x) = 0$ when $p_{\mathrm{ref}}(\tau \mid x) = 0$).

Now take the integrand $g_\theta(\tau) \triangleq -\log p_\theta(\tau \mid x)$ (extended-valued if needed). By a direct change of measure,

$$\mathbb{E}_{\tau \sim p^\star(\cdot \mid x)}\big[g_\theta(\tau)\big] = \sum_{\tau \in \mathcal{T}_x} p^\star(\tau \mid x)\, g_\theta(\tau) \tag{55}$$

$$= \sum_{\tau \in \mathcal{T}_x} p_{\mathrm{ref}}(\tau \mid x)\, \frac{p^\star(\tau \mid x)}{p_{\mathrm{ref}}(\tau \mid x)}\, g_\theta(\tau)$$

$$= \mathbb{E}_{\tau \sim p_{\mathrm{ref}}(\cdot \mid x)}\big[w(\tau \mid x)\, g_\theta(\tau)\big] \tag{56}$$

$$= \mathbb{E}_{\tau \sim p_{\mathrm{ref}}(\cdot \mid x)}\big[w(\tau \mid x)\big(-\log p_\theta(\tau \mid x)\big)\big], \tag{57}$$

where $w(\tau \mid x)$ is given by (54). Hence the two objectives in (17) are identical for every $\theta$, and therefore have the same set of minimizers. This proves Thm. 5. $\qquad\square$

## C.6. Proof of Prop. 6

*Proof.* Fix a prompt $x$. Let $\tau \sim p_{\mathrm{ref}}(\cdot \mid x)$ be a random trajectory with (random) effective length $L \equiv L(\tau) \in \{1, 2, \dots\}$. Define per-step losses $\ell_t(\tau) \triangleq -\log \pi_\theta(y_t \mid x_t)$ and the corresponding per-step gradients $\nabla \ell_t(\tau)$. Let $w(\tau) \geq 0$ be the importance weight, and define

$$g_{\mathrm{all}}(\tau) \triangleq w(\tau)\sum_{t=1}^{L}\nabla\ell_t(\tau), \qquad g_{\mathrm{term}}(\tau) \triangleq w(\tau)\nabla\ell_L(\tau), \qquad \Delta g(\tau) \triangleq g_{\mathrm{all}}(\tau) - g_{\mathrm{term}}(\tau) = w(\tau)\sum_{t=1}^{L-1}\nabla\ell_t(\tau). \tag{58}$$

Assume the almost-sure bounds

$$0 \leq w(\tau) \leq W_{\max}, \qquad \|\nabla\ell_t(\tau)\| \leq G_{\max} \ \text{for all valid } t. \tag{59}$$

**Bias bound.** Since $\mathbb{E}[g_{\mathrm{all}}] - \mathbb{E}[g_{\mathrm{term}}] = \mathbb{E}[\Delta g]$, it suffices to bound $\|\mathbb{E}[\Delta g]\|$. By Jensen's inequality and (58)–(59),

$$\big\|\mathbb{E}[\Delta g]\big\| \leq \mathbb{E}\big[\|\Delta g\|\big] = \mathbb{E}\Big[\Big\|w(\tau)\sum_{t=1}^{L-1}\nabla\ell_t(\tau)\Big\|\Big] \tag{60}$$

$$\leq \mathbb{E}\Big[w(\tau)\sum_{t=1}^{L-1}\|\nabla\ell_t(\tau)\|\Big]$$

$$\leq \mathbb{E}\big[W_{\max}\,(L-1)\,G_{\max}\big] \tag{61}$$

$$= W_{\max}G_{\max}\cdot\mathbb{E}\big[L-1\big], \tag{62}$$

which proves (18).

**Variance reduction.** For random vectors $a, b$ with finite second moments, define

$$\mathrm{Var}(a) \triangleq \mathbb{E}\big[\|a - \mathbb{E}[a]\|^2\big], \qquad \mathrm{Cov}(a,b) \triangleq \mathbb{E}[\langle a - \mathbb{E}[a],\, b - \mathbb{E}[b]\rangle].$$

Using $g_{\text{all}} = g_{\text{term}} + \Delta g$ and the polarization identity,

$$\begin{aligned}
\text{Var}(g_{\text{all}}) &= \mathbb{E}\Big[\big\|(g_{\text{term}} - \mathbb{E}[g_{\text{term}}]) + (\Delta g - \mathbb{E}[\Delta g])\big\|^2\Big] \\
&= \text{Var}(g_{\text{term}}) + \text{Var}(\Delta g) + 2\,\text{Cov}(g_{\text{term}}, \Delta g).
\end{aligned} \tag{63}$$

Under the stated condition

$$\text{Var}(\Delta g) + 2\,\text{Cov}(g_{\text{term}}, \Delta g) > 0,$$

(63) implies $\text{Var}(g_{\text{all}}) > \text{Var}(g_{\text{term}})$, i.e.,

$$\text{Var}(g_{\text{term}}) < \text{Var}(g_{\text{all}}),$$

which proves (20). $\qquad\square$

### C.7. Proof of Prop. 7

*Proof.* Fix a prompt $x$. Let $\tau_1, \ldots, \tau_K$ be i.i.d. samples from $p_{\text{ref}}(\cdot \mid x)$, and define

$$X_i \triangleq \exp\Big(\frac{R(\tau_i)}{\beta}\Big), \tag{64}$$

$$Z(x) \triangleq \mathbb{E}_{\tau \sim p_{\text{ref}}(\cdot \mid x)}\Big[\exp\big(\tfrac{R(\tau)}{\beta}\big)\Big], \tag{65}$$

$$\widehat{Z}(x) \triangleq \frac{1}{K}\sum_{i=1}^{K} X_i. \tag{66}$$

By the bounded-return assumption $R(\tau) \in [R_{\min}, R_{\max}]$, we have for every $i$,

$$m_\beta \triangleq \exp\Big(\frac{R_{\min}}{\beta}\Big) \;\le\; X_i \;\le\; \exp\Big(\frac{R_{\max}}{\beta}\Big) \triangleq M_\beta, \tag{67}$$

and hence the range satisfies $\Delta_\beta \triangleq M_\beta - m_\beta$.

Applying Hoeffding's inequality for independent bounded random variables yields, for any $\epsilon > 0$,

$$\mathbb{P}\Big(\big|\widehat{Z}(x) - Z(x)\big| \ge \epsilon\Big) \le 2\exp\Big(-\frac{2K^2\epsilon^2}{\sum_{i=1}^{K}(M_\beta - m_\beta)^2}\Big). \tag{68}$$

Moreover,

$$\sum_{i=1}^{K}(M_\beta - m_\beta)^2 = \sum_{i=1}^{K}\Delta_\beta^2 \tag{69}$$

$$= K\Delta_\beta^2, \tag{70}$$

and therefore

$$\mathbb{P}\Big(\big|\widehat{Z}(x) - Z(x)\big| \ge \epsilon\Big) \le 2\exp\Big(-\frac{2K\epsilon^2}{\Delta_\beta^2}\Big), \tag{71}$$

which proves (22).

To obtain (23), it suffices to enforce

$$2\exp\Big(-\frac{2K\epsilon^2}{\Delta_\beta^2}\Big) \le \delta. \tag{72}$$

Taking logarithms and rearranging,

$$-\frac{2K\epsilon^2}{\Delta_\beta^2} \le \log\Big(\frac{\delta}{2}\Big), \tag{73}$$

$$\frac{2K\epsilon^2}{\Delta_\beta^2} \ge \log\Big(\frac{2}{\delta}\Big), \tag{74}$$

$$K \ge \frac{\Delta_\beta^2}{2\epsilon^2}\log\Big(\frac{2}{\delta}\Big), \tag{75}$$

which is exactly (23). □

# D. Experiments

## D.1. Baseline

**SFT-5TURN.** We fine-tune the model with SFT on the same offline correction trajectories as DRIFT, with up to five turns per example. Since in our setting only the final-turn response can be correct, we follow DRIFT and train only on the final-turn response conditioned on the full preceding dialogue context. This baseline is equivalent to DRIFT with all trajectory weights set to 1, and thus serves as the unweighted ablation.

**STAR-2TURN.** We implement STaR-2Turn (Zelikman et al., 2022) as a two-turn self-training baseline adapted to our correction protocol. For each training prompt, we first sample a turn-1 response from the current model and evaluate it using the same verifier as in our main setup. If the turn-1 response is incorrect, we append the fixed lightweight negative feedback and prompt the model to produce a turn-2 revision. We keep only those trajectories whose turn-2 response is verified as correct, and then fine-tune the model with SFT on the turn-2 responses conditioned on the full two-turn context. Under our setting, only the final-turn response can be correct, so we do not fit the turn-1 responses. This yields an implementation of STaR that bootstraps training data from verified two-turn corrections.

**SCORE-2TURN.** SCoRe (Kumar et al., 2024) studies learning from corrective feedback with verifiable rewards in a two-attempt setting. The model produces an initial answer, receives lightweight negative feedback if it is incorrect, and then generates a second-turn revision. Training uses KL regularization to control deviation from a reference policy and encourages successful correction from an incorrect first attempt to a correct second attempt. In our comparisons, we treat SCoRe as a two-turn training approach under this fixed correction protocol.

**UFO-5TURN.** UFO (Liu et al., 2025) considers multi-turn trial-and-error prompting and training under generic negative feedback. The model repeatedly attempts the same problem, and after each incorrect attempt it receives a fixed message such as "Incorrect, please try again" before producing the next response. The process stops once a correct answer is obtained or a maximum turn budget is reached. In our setting, we refer to UFO with a maximum of five turns as UFO-5TURN.

## D.2. Setup

During the offline trajectory generation phase, we set the number of rollouts to $K = 16$ with a sampling temperature of 1.0. The maximum trajectory length is restricted to $T = 5$, and the maximum number of new tokens is set to 512. In the optimization phase, we utilize a global batch size of 128 and train the model for 200 steps. The hyperparameters are configured as $\beta = 0.1$ and $\gamma = 0.9$, and we set the repetition penalty coefficient to $\lambda = 0.5$.

## D.3. Benchmark

We evaluate multi-turn correction on benchmarks spanning mathematical reasoning and general-domain knowledge and science reasoning. Because training uses only the MATH subset of MetaMathQA, we treat the mathematical reasoning benchmarks as same-family evaluations and the general reasoning benchmarks as out-of-domain evaluations that test whether correction behavior transfers beyond the training task. Table 2 summarizes the benchmark scale.

*Table 2.* Benchmarks used for evaluation. "N/A" indicates that the benchmark is used only for evaluation in our setting and we do not use a predefined training split from that dataset.

| Domain | Benchmark | Training Size | Test Size |
|---|---|---|---|
| Mathematical Reasoning | MATH | 7,500 | 5,000 |
| | MATH500 | N/A | 500 |
| | TheoremQA | N/A | 800 |
| General Reasoning | MMLU-Redux | N/A | 5,700 |
| | MMLU-Pro | N/A | 12,032 |
| | GPQA-diamond | N/A | 198 |

**Mathematical reasoning.** We use MATH (Hendrycks et al., 2021) as the primary math benchmark with competition-style problems that require multi-step derivations. We also report MATH500 (Hendrycks et al., 2021), a 500-problem evaluation

*Table 3.* Additional model results on Qwen2.5-7B-Instruct after training on MetaMathQA (MATH subset). We report **multi@5** accuracy (%) with a maximum budget of 5 turns. Deltas are computed w.r.t. the base model.

| Model | Method | Math | | | | General | | | | All |
|---|---|---|---|---|---|---|---|---|---|---|
| | | MATH | MATH500 | ThmQA | Avg | MMLU-R | MMLU-P | GPQA | Avg | Avg |
| Qwen2.5-7B -Instruct | Base | 62.2 | 64.0 | 37.1 | 54.4 | 90.5 | 68.9 | 66.1 | 75.2 | 64.8 |
| | SFT-5turn | 64.2 ↑2.0 | 64.4 ↑0.4 | 37.2 ↑0.1 | 55.3 ↑0.9 | 90.5 →0.0 | 69.3 ↑0.4 | 68.1 ↑2.0 | 76.0 ↑0.8 | 65.6 ↑0.8 |
| | UFO-5turn | 66.3 ↑4.1 | 66.2 ↑2.2 | **38.6** ↑1.5 | 57.0 ↑2.6 | **92.3** ↑1.8 | 70.6 ↑1.7 | **73.2** ↑7.1 | **78.7** ↑3.5 | 67.9 ↑3.1 |
| | DRIFT-5turn | **67.6** ↑5.4 | **68.6** ↑4.6 | 38.5 ↑1.4 | **58.2** ↑3.8 | 91.2 ↑0.7 | **71.2** ↑2.3 | 72.7 ↑6.6 | 78.4 ↑3.2 | **68.3** ↑3.5 |

*Table 4.* A simple rollout-refresh variant on Qwen2.5-3B-Instruct. We report **multi@5** accuracy (%) with a maximum budget of 5 turns. Deltas are computed w.r.t. the base model, and both DRIFT variants use the same total optimization budget.

| Model | Method | Math | | | | General | | | | All |
|---|---|---|---|---|---|---|---|---|---|---|
| | | MATH | MATH500 | ThmQA | Avg | MMLU-R | MMLU-P | GPQA | Avg | Avg |
| Qwen2.5-3B -Instruct | Base | 38.3 | 40.2 | 26.0 | 34.8 | 76.8 | 49.0 | 47.9 | 57.9 | 46.4 |
| | DRIFT-5turn | 55.9 ↑17.6 | 58.2 ↑18.0 | 34.3 ↑8.3 | 49.5 ↑14.7 | 84.6 ↑7.8 | 57.2 ↑8.2 | **72.7** ↑24.8 | 71.5 ↑13.6 | 60.5 ↑14.1 |
| | +refresh | **56.7** ↑18.4 | **59.6** ↑19.4 | **34.8** ↑8.8 | **50.4** ↑15.6 | **85.9** ↑9.1 | **57.7** ↑8.7 | **72.7** ↑24.8 | **72.1** ↑14.2 | **61.2** ↑14.8 |

subset commonly used for faster iteration. In addition, we include TheoremQA (ThmQA) (Chen et al., 2023), which tests theorem-driven problem solving on STEM questions and emphasizes selecting and applying appropriate theorems.

**General reasoning.** To evaluate cross-domain transfer, we include MMLU-Redux (MMLU-R) (Gema et al., 2025), a cleaned and re-annotated version of MMLU designed to reduce ambiguity and labeling errors. We further evaluate on MMLU-Pro (MMLU-P) (Wang et al., 2024), which increases difficulty by expanding answer options and filtering trivial items. Finally, we use GPQA-diamond (Rein et al., 2023), a high-quality and challenging subset of GPQA that focuses on graduate-level science questions.

## D.4. Additional Model Results

To further evaluate whether DRIFT scales to a stronger backbone, we additionally train and evaluate Qwen2.5-7B-Instruct under the same protocol. Table 3 shows that DRIFT improves the all-benchmark average from 64.8% to 68.3%, and slightly outperforms the online multi-turn RL baseline while retaining the largest gains on MATH and MATH500.

## D.5. A Simple Rollout-Refresh Variant

A natural limitation of a single offline rollout collection is coverage: if the reference policy does not produce a useful correction trajectory, importance weighting cannot recover it during optimization. To test whether this limitation can be partially mitigated without fully reverting to online RL, we run a simple two-stage rollout-refresh variant on Qwen2.5-3B-Instruct. We first train DRIFT for 100 steps, use the resulting checkpoint to regenerate correction trajectories, and then continue DRIFT training for another 100 steps. The total optimization budget is kept the same as the single-stage 200-step DRIFT run.

Table 4 shows that this simple refresh variant improves the all-benchmark average from 60.5% to 61.2%. The gain is modest, but it suggests that periodically refreshing the rollout distribution can partially address the coverage limitation of one-shot offline rollouts. This result is best interpreted as an initial diagnostic rather than a complete replacement for online exploration; designing more systematic rollout-refresh schedules is left for future work.

## D.6. DRIFT as an Initialization for Online RL

We further examine whether DRIFT can be used as an initialization before online multi-turn RL. This experiment uses Qwen2.5-3B-Instruct and keeps the total optimization budget fixed at 200 steps. Pure UFO and pure DRIFT are trained for 200 steps, while the hybrid schedules use the first 100 steps for either SFT or DRIFT and the remaining 100 steps for UFO.

Table 5 reports all improvements relative to the base model. DRIFT followed by UFO achieves the best all-benchmark average among these schedules and improves over pure UFO by 2.3 points. In contrast, SFT followed by UFO does not improve over pure UFO, suggesting that the benefit is not simply due to adding an offline warm-up stage. These results

*Table 5.* DRIFT as an initialization for online multi-turn RL on Qwen2.5-3B-Instruct. We report **multi@5** accuracy (%) with a maximum budget of 5 turns. Deltas are computed w.r.t. the base model.

| Model | Method | Math | | | | General | | | | All |
|---|---|---|---|---|---|---|---|---|---|---|
| | | MATH | MATH500 | ThmQA | Avg | MMLU-R | MMLU-P | GPQA | Avg | Avg |
| Qwen2.5-3B -Instruct | Base | 38.3 | 40.2 | 26.0 | 34.8 | 76.8 | 49.0 | 47.9 | 57.9 | 46.4 |
| | UFO-5turn | 55.5 $_{\uparrow17.2}$ | 56.4 $_{\uparrow16.2}$ | 33.5 $_{\uparrow7.5}$ | 48.5 $_{\uparrow13.7}$ | 87.0 $_{\uparrow10.2}$ | 57.1 $_{\uparrow8.1}$ | 71.2 $_{\uparrow23.3}$ | 71.8 $_{\uparrow13.9}$ | 60.2 $_{\uparrow13.8}$ |
| | DRIFT-5turn | 55.9 $_{\uparrow17.6}$ | 58.2 $_{\uparrow18.0}$ | 34.3 $_{\uparrow8.3}$ | 49.5 $_{\uparrow14.7}$ | 84.6 $_{\uparrow7.8}$ | 57.2 $_{\uparrow8.2}$ | **72.7** $_{\uparrow24.8}$ | 71.5 $_{\uparrow13.6}$ | 60.5 $_{\uparrow14.1}$ |
| | SFT+UFO | 58.3 $_{\uparrow20.0}$ | 57.4 $_{\uparrow17.2}$ | 34.5 $_{\uparrow8.5}$ | 50.1 $_{\uparrow15.3}$ | 82.1 $_{\uparrow5.3}$ | 55.6 $_{\uparrow6.6}$ | 62.1 $_{\uparrow14.2}$ | 66.6 $_{\uparrow8.7}$ | 58.3 $_{\uparrow11.9}$ |
| | DRIFT+UFO | **60.3** $_{\uparrow22.0}$ | **61.4** $_{\uparrow21.2}$ | **34.8** $_{\uparrow8.8}$ | **52.2** $_{\uparrow17.4}$ | **87.6** $_{\uparrow10.8}$ | **58.5** $_{\uparrow9.5}$ | 72.2 $_{\uparrow24.3}$ | **72.8** $_{\uparrow14.9}$ | **62.5** $_{\uparrow16.1}$ |

provide preliminary evidence that DRIFT can serve as a useful warm start for online RL in this verifier-guided setting, although a broader study of hybrid training schedules is left for future work.

## D.7. Prompt

In this section, we present the prompts used for rollout and evaluation.

---

**Rollout for MetaMathQA**

<system>
You're a helpful assistant.
</system>
<user>
You are solving Math problems.  Only give the final answer between <answer> and
</answer>.
Turn 1:
State:  **[Problem]**
You have 5 actions left.  Always output:  <think> [Your thoughts] </think> <answer>
[your answer] </answer> with no extra text.  Strictly follow this format.  Max response
length:  400 words (tokens).
</user>

---

**Evaluation for MATH / MATH500**

<system>
You're a helpful assistant.
</system>
<user>
You are solving Math problems.  Only give the final answer between <answer> and
</answer>.
Problem:  **[Problem]**
Always output:  <think> [Your thoughts] </think> <answer> [your answer] </answer> with
no extra text.  Strictly follow this format.
</user>

---

**Evaluation for TheoremQA**

<system>
You're a helpful assistant.
</system>
<user>
You are solving Math problems.  Only give the final answer between <answer> and
</answer>.
Problem:  <image>
**[Problem]**
Always output:  <think> [Your thoughts] </think> <answer> [your answer] </answer> with
no extra text.  Strictly follow this format.
</user>

---

---

**Evaluation for MMLU-Redux / MMLU-Pro**

<**system**>
```
You're a helpful assistant.
```
</**system**>
<**user**>
```
You are solving multiple-choice questions.  Only give the final answer letter (A-J)
between <answer> and </answer>.
```
Problem:  **[Problem]**
```
Always output:  <think> [Your thoughts] </think> <answer> [your answer] </answer> with
no extra text.  Strictly follow this format.
```
</**user**>

---

**Evaluation for GPQA**

<**system**>
```
You're a helpful assistant.
```
</**system**>
<**user**>
```
You are solving multiple-choice questions.  Only give the final answer letter (A-D)
between <answer> and </answer>.
```
Problem:  **[Problem]**
```
Always output:  <think> [Your thoughts] </think> <answer> [your answer] </answer> with
no extra text.  Strictly follow this format.
```
</**user**>

---

### D.8. Ablation on return shaping

We add a penalty term of the form $\lambda \left( 1 - \frac{E(\tau)}{L} \right)$ to the trajectory return to encourage generating diverse answers. We report the ablation results in Table 6 and Table 7. After removing the trajectory penalty term, the model exhibits decreases in accuracy, correction rate, and the average number of unique answers. This indicates that the trajectory penalty encourages the model to change its responses when it is wrong, which in turn leads to improved accuracy.

*Table 6.* Reward shaping ablation (5-turn budget; $N$=500). We report first-turn accuracy (Acc@1), cumulative 5-turn accuracy (Acc@5), and the correction rate relative to turn 1: $\text{Corr} = \frac{\text{Acc@5} - \text{Acc@1}}{1 - \text{Acc@1}}$. We also report the average number of unique answers on incorrect trajectories, and the overall average number of unique answers. Deltas are computed w.r.t. DRIFT.

| Method | Acc@1 (%) | Acc@5 (%) | Corr. (%) ↑ | Avg. #Unique (wrong traj.) | Avg. #Unique (overall) |
|---|---|---|---|---|---|
| DRIFT | 38.6 | **58.2** | **31.9** | **2.038** | **1.666** |
| - trajectory penalty | 36.0 $_{\downarrow 2.6}$ | 55.8 $_{\downarrow 2.4}$ | 30.9 $_{\downarrow 1.0}$ | 1.760 $_{\downarrow 0.278}$ | 1.560 $_{\downarrow 0.106}$ |

*Table 7.* Reward shaping ablation on cross-benchmark generalization after training on MetaMathQA (MATH subset). We report **multi@5** accuracy (%) with a maximum budget of 5 turns. Deltas are computed w.r.t. DRIFT-5turn.

| Method | Math | | | | General | | | | All |
| | MATH | MATH500 | ThmQA | Avg | MMLU-R | MMLU-P | GPQA | Avg | Avg |
|---|---|---|---|---|---|---|---|---|---|
| DRIFT | 55.9 | 58.2 | 34.3 | 49.5 | 84.6 | 57.2 | 72.7 | 71.5 | 60.5 |
| - trajectory penalty | 53.4 $_{\downarrow 2.5}$ | 55.8 $_{\downarrow 2.4}$ | 32.6 $_{\downarrow 1.7}$ | 47.3 $_{\downarrow 2.2}$ | 82.3 $_{\downarrow 2.3}$ | 56.1 $_{\downarrow 1.1}$ | 67.5 $_{\downarrow 5.2}$ | 68.6 $_{\downarrow 2.9}$ | 58.0 $_{\downarrow 2.5}$ |

### D.9. Learning Curves Across Hyperparameter Settings

We examine DRIFT's training dynamics under different choices of the discount factor $\gamma$ in the shaped trajectory return and the parameter $\beta$ in the exponential trajectory reweighting with prompt-level normalization. Figures 7 and 8 report multi-turn accuracy over training steps. Across a broad range of $\gamma$ and $\beta$, the learning curves improve steadily and remain stable throughout training, without abrupt collapse or divergence. While different settings lead to modest differences in the final accuracy and the amount of fluctuation, the overall trend is consistent, suggesting that DRIFT is not brittle to these hyperparameters in practice.

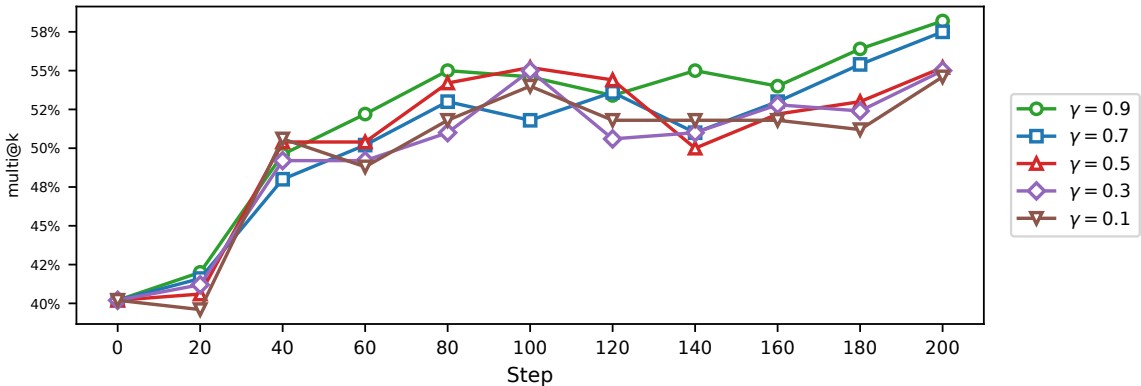

*Figure 7.* Accuracy under different $\gamma$ values over training steps.

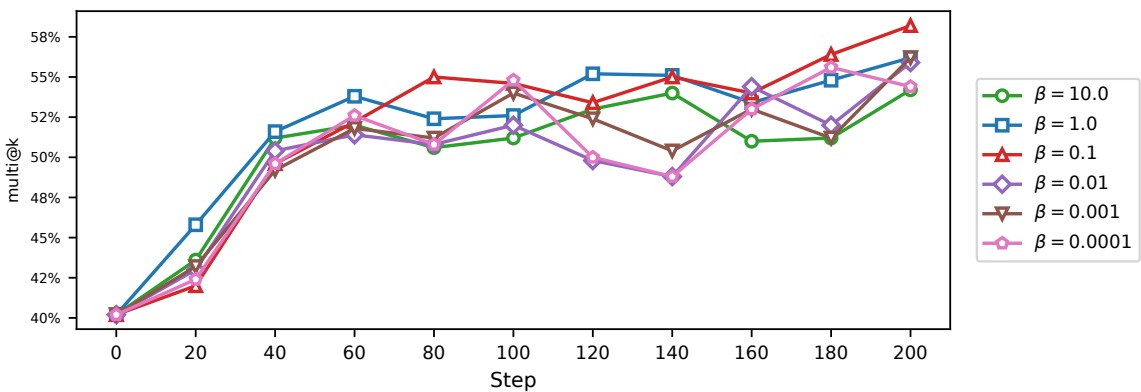

*Figure 8.* Accuracy under different $\beta$ values over training steps.

We plot the training curves under different rollout numbers $K$ in Fig. 9 while keeping the batch size and training steps fixed. Table 8 summarizes the group-level data distribution for different $K$. The results show that when $K$ is small, groups are more likely to become degenerate (all-correct or all-wrong), increasing the proportion of such groups. In this regime, the method gradually approaches an SFT-like update with nearly uniform (all-one) weights, and the performance becomes close to SFT-5Turn, leading to worse-than-expected results. In contrast, larger rollouts typically yield better performance.

## E. Case Study

### E.1. Case 1

We qualitatively analyze a failure case from MATH-500 in Figure 10 to demonstrate DRIFT's error-recovery capability. The problem requires finding integer values of $x$ where $x^3 | 10!$ (implying the exponent constraint $3c \le 2$ for prime factor 5).

As shown in Figure 10, the base model suffers from behavioral collapse, repeating the logically inconsistent claim that "$c$ can be 0 or 1" (despite $3(1) > 2$) across four turns. Generic feedback fails to break this local optimum. In contrast, DRIFT effectively leverages negative feedback to prune the search space. In Turn 3, it explicitly re-verifies that $5^3 \nmid 10!$, correcting the exponent count to 1. This illustrates that DRIFT learns to explore alternative reasoning paths rather than merely resampling high-probability errors.

*Table 8.* Group composition and trajectory success rate under different rollout numbers $K$ (batch size and training steps fixed). "Effective" groups are mixed groups (neither all-correct nor all-wrong).

| $K$ | All-correct (%) | All-wrong (%) | Effective/Mixed (%) | Accuracy (%) |
|---|---|---|---|---|
| 4 | 52.9 | 11.5 | 35.5 | 73.1 |
| 8 | 42.3 | 7.8 | 49.9 | 73.4 |
| 16 | 34.6 | 5.8 | 59.6 | 73.9 |
| 32 | 25.0 | 3.8 | 71.2 | 74.6 |
| 64 | 21.2 | 2.5 | 76.3 | 75.0 |
| 128 | 12.7 | 2.0 | 85.3 | 74.5 |

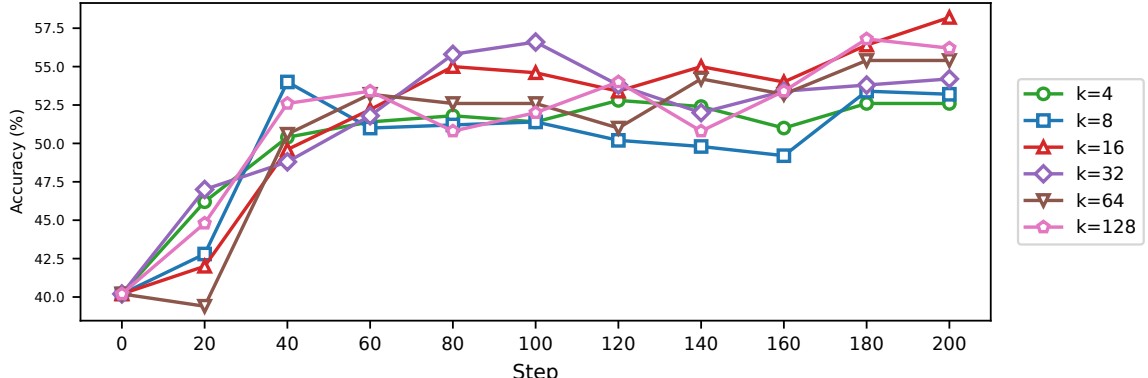

*Figure 9.* Accuracy under different $K$ values over training steps.

### E.2. Case 2

In Figure 11, we examine a modular arithmetic problem from MATH-500. This case illustrates a different failure mode: hallucinated verification. The base model incorrectly verifies a wrong answer and becomes stuck in a loop because it cannot correct its own arithmetic error despite external feedback. DRIFT, conversely, successfully explores different candidate solutions across turns.

### E.3. Case 3: Limitations in Knowledge-Intensive Domains

he primary contribution of DRIFT is to enable verifiable multi-turn optimization with the training efficiency of SFT. While DRIFT demonstrates strong performance on reasoning-intensive tasks (e.g., MATH) by effectively learning correction policies, we analyze an out-of-distribution case in Figure 12 to distinguish between learning *how to correct* (strategy) and possessing the *domain knowledge* required to correct (capability).

In this organic chemistry problem from GPQA:

- **Base Model:** It selects the correct answer (C) in Turn 1, but relies on a hallucinated mechanism ("electrophilic aromatic substitution") that is chemically impossible for the given aliphatic structure.

- **UFO:** Lacking the knowledge to verify the answer, it falls into an exhaustive guessing loop, repeating the exact same reasoning text while cycling through options (B → A → B → D → C).

- **DRIFT:** Its reasoning ("forming a carbocation") remains heuristically plausible but factually flawed for the concerted mCPBA mechanism.

This comparison highlights that DRIFT successfully instills the strategic behavior of multi-turn correction, allowing the model to avoid the blind trial-and-error loops seen in baselines like UFO. However, the accuracy of the reasoning is bounded by the model's intrinsic knowledge. Since DRIFT focuses on the optimization framework for efficiency rather than domain

| Problem: If $x^3$ is a positive factor of 10!, how many possible integer values of $x$ are there? (Reminder: $10! = 2^8 \cdot 3^4 \cdot 5^2 \cdot 7$) | |
|---|---|
| **Base Model (Qwen2.5-3B-Instruct)** | **Ours (DRIFT)** |
| **Turn 1**  ✗**Incorrect (18)**
"... For $x^3$ to be a factor of 10!... $3c \le 2$. This means $c$ can be 0 or 1 (2 choices)... Total... $3 \times 3 \times 2 \times 1 = 18$." | **Turn 1**  ✗**Incorrect (10)**
"... This means that $x$ must be composed of the prime factors of 10! in such a way that when cubed, it still divides 10!... [General conceptual error]" |
| *User: Incorrect. Please think again.* | *User: Incorrect. Please think again.* |
| **Turn 2**  ✗**Incorrect (12)**
"... $3c \le 2$, **so $c$ can be 0 or 1 (2 choices)**... The total number... is $3 \times 2 \times 2 \times 1 = 12$." | **Turn 2**  ✗**Incorrect (30)**
"... For prime factor 5, possible exponents in $x$ are 0, 1 (since $5^3 = 125$... not a factor of $5^2$). **This gives us 2 possibilities.** ... Total... 30." |
| *User: Incorrect. Please think again.* | *User: Incorrect. Please think again.* |
| **Turn 3**  ✗**Incorrect (12)**
"... $3c \le 2$, **so $c$ can be 0 or 1 (2 choices)**... The total number... is $3 \times 2 \times 2 \times 1 = 12$." | **Turn 3**  ✓**Correct (6)**
"... For prime factor 5, possible exponents in $x$ are 0 (since $5^3 = 125$... not a factor of $5^2$). **This gives us 1 possibility.** ... Total... $3 \times 2 \times 1 \times 1 = 6$." |
| *User: Incorrect. Please think again.*
**Turn 4 & 5**  ✗**Incorrect (12)**
*[Repeats identical erroneous logic as Turn 2 & 3]* | |

*Figure 10.* Trajectory comparison. The Base Model (Qwen2.5-3B-Instruct) correctly sets up the inequality $3c \le 2$ but falls into a reasoning loop, repeatedly miscalculating the integer solutions despite feedback. DRIFT initially errs but successfully uses feedback in Turn 3 to correct its analysis of the prime factor 5 ($5^3 \nmid 10!$), deriving the correct answer.

adaptation, it does not spontaneously inject missing subject matter expertise. This is not a limitation of the DRIFT objective itself, but rather indicates that for knowledge-intensive domains, the learned correction policy must be supported by sufficient underlying capabilities. A promising direction for future work is to couple the efficient correction training of DRIFT with domain-specific continual pre-training, ensuring the model possesses both the policy to correct and the knowledge to correct accurately.

**Problem:** Find the unique **odd** integer $t$ such that $0 < t < 23$ and $t + 2$ is the inverse of $t$ modulo 23.

| Base Model (Qwen2.5-3B-Instruct) | | Ours (DRIFT) | |
|---|---|---|---|
| **Turn 1** | ✗**Incorrect (5)** | **Turn 1** | ✗**Incorrect (5)** |
| "... We need to find $t$ such that $t^2 + 2t - 1 \equiv 0 \pmod{23}$. We need to check each odd integer from 1 to 21..." | | "... This implies $(t + 2) \cdot t \equiv 1 \pmod{23}$. We need to test each odd integer between 1 and 21..." | |
| *User: Incorrect. Please think again.* | | *User: Incorrect. Please think again.* | |
| **Turn 2** | ✗**Incorrect (19)** | **Turn 2** | ✗**Incorrect (15)** |
| "Let's test the odd integers... 
 - For $t = 17$: $17^2 + 34 - 1 = 322 \equiv 16 \not\equiv 0$ 
 - For $t = 19$: $19^2 + 38 - 1 = 400 \equiv 0 \pmod{23}$ 
 We find that $t = 19$ satisfies the equation..." 
 *[Note: $400 \equiv 9 \pmod{23}$, not 0. Hallucinated calculation.]* | | "Given the previous attempt was incorrect, let's re-evaluate... We will test each odd integer... [Model attempts a different candidate in the search space]." | |
| *User: Incorrect. Please think again.* | | *User: Incorrect. Please think again.* | |
| **Turn 3** | ✗**Incorrect (19)** | **Turn 3** | ✓**Correct (17)** |
| "Let's test the odd integers... 
 - For $t = 19$: $19^2 + 38 - 1 = 400 \equiv 0 \pmod{23}$ 
 Therefore, the unique odd integer $t$ is 19." | | "... We will test each odd integer... [Implicitly verifies 17]. 
 *[Note: $17^2 + 2(17) - 1 = 322 = 14 \times 23 \equiv 0$.]*" | |
| *User: Incorrect. Please think again.* | | | |
| **Turn 4 & 5** | ✗**Incorrect (19)** | | |
| *[Repeats exact same calculation error and output]* | | | |

*Figure 11.* Trajectory comparison on a modular arithmetic problem. The Base Model commits a calculation error in Turn 2, falsely believing that 400 is divisible by 23. It becomes confident in this incorrect verification and ignores subsequent negative feedback, repeating the answer 19. DRIFT treats the feedback as a signal to explore the solution space, moving from candidate 5 to 15, and finally verifying 17 correctly in Turn 3.

**Problem (GPQA):** Reaction of 3,3,6-trimethylhepta-1,5-dien-4-one with mCPBA followed by Methyllithium/CuI. (Requires knowledge of epoxidation and Gilman reagent mechanisms)

| Base Model | UFO (RL Baseline) | Ours (DRIFT) |
|---|---|---|
| **Turn 1** ✓**Correct (C)** | **Turn 1** ✗**Incorrect (B)** | **Turn 1** ✓**Correct (C)** |
| *Reasoning:* "...leads to diol derivative due to **electrophilic aromatic substitution**..." 

 *[Analysis: Factually wrong mechanism. Lucky guess.]* | *Reasoning:* "...undergo epoxidation... form tertiary alcohol... product is 5-hydroxy..." 
 *[Analysis: Plausible start, wrong conclusion.]* | *Reasoning:* "...cleavage of the double bond... **forming a carbocation**..." 
 *[Analysis: Factually wrong mechanism. Heuristic guess.]* |
| | **Turn 2** ✗**Incorrect (A)** 
 *Reasoning:* [Exact same text as Turn 1] "...product is 2,3,4,5,5-..." | |
| | **Turn 3** ✗**Incorrect (B)** 
 *Reasoning:* [Exact same text as Turn 1] "...product is 5-hydroxy..." | |
| | **Turn 4** ✗**Incorrect (D)** 
 *Reasoning:* [Exact same text as Turn 1] "...product is 4,4,5,7,7-..." | |
| | **Turn 5** ✓**Correct (C)** 
 *Reasoning:* [Exact same text as Turn 1] "...product is 6-hydroxy..." 
 *[Analysis: Blind exhaustive guessing.]* | |

*Figure 12.* Risk Analysis on GPQA. UFO exposes the risk of blind guessing by cycling through options. Critically, the Base model and DRIFT also resort to guessing, relying on hallucinated mechanisms (e.g., "carbocation") despite selecting the correct option. This collective failure in reasoning reveals a fundamental capability deficit: without domain knowledge, models revert to various forms of guessing rather than genuine problem-solving, regardless of the optimization strategy.

