# OpenReview forum: "DRIFT: Decoupled Rollouts and Importance-Weighted Fine-Tuning for Efficient Multi-Turn Optimization"
_ICML.cc/2026/Conference — ICML 2026 regular_

### Official Review · Reviewer_ZhG2 · 2026-03-11

**Soundness:** 4
**Presentation:** 4
**Significance:** 3
**Originality:** 3
**Overall Recommendation:** 5
**Confidence:** 4

**Summary:**

The authors address efficient multi-turn self-correction training for LLMs receiving binary feedback. They show the KL-regularized RL setting can be reformulated as importance weighted SFT over offline trajectories, allowing rollout generation and policy optimization to be done separately. Training on only the final correctly generated responses on math and reasoning benchmarks can match online RL baselines with compute efficiency comparable to SFT training.

**Compliance With Llm Reviewing Policy:**

Affirmed.

**Final Justification:**

Rebuttal addressed my main concerns, but my score was relatively high compared to the other reviewers. Additionally, I feel to give a 6 the paper must be exceptional, and go beyond being a solid piece of research. Because of this, I maintain my score of 5.

**Key Questions For Authors:**

1. How restrictive / true-in-practice is the assumption at the start of Lemma 3?
2. Have you searched through the offline/batched/entropy-regularised RL literature for theoretical results similar to the one that motivated your method? What comes closest?

**Limitations:**

Limitations are not discussed.

**Strengths And Weaknesses:**

### Strengths
1. The theorems and theoretical analysis are nice and well-motivate the method
2. The exposition is clear and the ideas and methods are presented with good clarity
3. Experiments are rigorous and cover two different models from different families, include many baselines, and cover many datasets
4. Empirical results seem strong and show the method works well, both in performance and compute costs
5. Ablations are solid and justify design choices as well as confirming predictions from the theory

### Weaknesses
1. The theory and method does not acknowledge that a key weakness of DRIFT vs RL is that since it does not iteratively sample and train, it cannot explore novel trajectories / methods. It is at most as strong as a single step of RL with a similar batch size. This is especially pertinent as the number of samples from the reference policy is finite and very small with respect to the space of possible completions, so it is unlikely to have sampled and up-weighted highly effective solutions that RL could find over many iterations. I do not think this is a critical weakness, but it should be acknowledged.
2. I'm worried that prior work exists in the offline/batched/entropy-regularised RL literature that might have similar theoretical insights/contributions, lowering the originality of this work, but I'm not familiar in-depth with these fields and they are not discussed in the paper. (Although, even if this were the case, the paper would still be a novel application of these theoretical insights.)

---

> ### Author Rebuttal · Authors · 2026-03-30
>
> We sincerely thank the reviewer for recognizing and positively evaluating the theoretical analysis, methodological motivation, clarity, and empirical results of our work. We have carefully considered your questions and suggestions, provided our responses, and will incorporate them into the revised version. If you have any further concerns, please contact us for discussion.
>
> # W1: DRIFT vs RL
>
> Thank you for this valuable suggestion. We agree with your point that, because DRIFT samples only once and does not alternate rollout and optimization iteratively, it may fail to explore solutions that online RL could discover through repeated iterations. We will explicitly include this as a limitation in the revised version. We also experimented with a simple scaling-based mitigation strategy as a possible remedy: instead of performing a single 200-step training run, we split it into two stages. After the first 100 training steps, we use the resulting checkpoint as a new starting point, regenerate rollouts, and continue training. This iterative variant shows better performance than the non-iterative DRIFT. The results are as follows:
>
> | Model               | Method                  | MATH | MATH500 | ThmQA | MMLU-R | MMLU-P | GPQA |  All Avg |
> | ------------------- | ----------------------- | ---: | ------: | ----: | -----: | -----: | ---: | -------: |
> | Qwen2.5-3B-Instruct | DRIFT-5turn             | 55.9 |    58.2 |  34.3 |   84.6 |   57.2 | 72.7 |     60.5 |
> |                     | DRIFT-5turn (two stage) | 56.7 |    59.6 |  34.8 |   85.9 |   57.7 | 72.7 | **61.2** |
>
>
>
> # W2 & Q2: Related Work
>
> Thank you for this valuable question. There is indeed a clear prior theoretical lineage connecting regularized RL and reward-weighted likelihood optimization, and this should have been discussed more explicitly in the main paper rather than only in the appendix. In related-work section of appendix A, we cite the broader KL-/entropy-regularized RL and reward-/advantage-weighted regression line, as well as recent reweighted-SFT works like **Refit[1]** and **VAR[2]**. Among these papers, DRIFT appliesthese KL-regularized, weighted-likelihood principles at the trajectory level using offline rollouts under a reference policy,prompt-level weight normalization, and terminal-only supervision, decoupling rollout from optimization and achieving anRL-motivated target with SFT-like efficiency.
>
> [1] Offline RL by Reward-Weighted Fine-Tuning for Conversation Optimization
>
> [2] RLHF in an SFT Way: From Optimal Solution to Reward-Weighted Alignment
>
>
>
> # Q1: Assumption of Lemma3
>
> Thank you for this important question. We agree that the realizability assumption in Lemma 3 is strong if interpreted literally as requiring $p^\star \in \Pi_\theta$. Its role is specifically to justify replacing the reverse-KL target by the forward-KL / weighted-SFT surrogate at the level of global optima.
>
> In practice, however, we view the more relevant regime as local rather than global. In our setting,
> $$
> p^\star(\tau \mid x)\propto p_{\mathrm{ref}}(\tau \mid x)\exp(R(\tau)/\beta),
> $$
> so $p^\star$ is not an arbitrary target distribution but an exponential tilting of the reference policy. Therefore, for bounded rewards and moderate $\beta$, the target remains in a neighborhood of $p_{\mathrm{ref}}$, which is also where optimization starts.
>
> More formally, under standard shared-support regularity, let
> $$
> \Delta_x := \mathrm{TV}(p_\theta(\cdot\mid x),\, p^\star(\cdot\mid x)).
> $$
> Then a Taylor expansion of both divergences around $p_\theta = p^\star$ gives
> $$
> \mathrm{KL}(p_\theta\|p^\star)
> = \frac{1}{2}\sum_{\tau}\frac{(p_\theta(\tau\mid x)-p^\star(\tau\mid x))^2}{p^\star(\tau\mid x)} + O(\Delta_x^3),
> $$
> and
> $$
> \mathrm{KL}(p^\star\|p_\theta)
> = \frac{1}{2}\sum_{\tau}\frac{(p_\theta(\tau\mid x)-p^\star(\tau\mid x))^2}{p^\star(\tau\mid x)} + O(\Delta_x^3).
> $$
> Therefore,
> $$
> \big|\mathrm{KL}(p_\theta\|p^\star)-\mathrm{KL}(p^\star\|p_\theta)\big|
> \le C_x \Delta_x^3
> $$
> for some constant $C_x$. Hence, even when exact realizability does not hold, the weighted-SFT objective remains a locally accurate surrogate for the original KL-regularized RL objective once $p_\theta$ is close to $p^\star$. Since $p_\theta$ is initialized from $p_{\mathrm{ref}}$ and optimized under the same KL-regularized objective, the practically relevant regime is precisely one where both $p_\theta$ and $p^\star$ stay in a neighborhood of $p_{\mathrm{ref}}$, making $\Delta_x$ small. Therefore, while realizability is needed for an exact global equivalence claim, it is not needed for the practical local validity of our surrogate objective.
>
> This interpretation is also consistent with our empirical findings: very small $\beta$ leads to overly concentrated weights and worse stability, while a moderate $\beta$ performs best. This matches the picture that DRIFT operates in a near-reference, moderately tilted regime. We will clarify this point in the revision and include the complete argument in the appendix.

---

> > ### Author Rebuttal · Reviewer_ZhG2 · 2026-04-03
> >
> > Thank you for addressing my concerns. Given the other reviews and the (lack of) granularity of the scoring system I will have to maintain my score. I think this is good paper but it doesn't quite go above and beyond to the extent implied by a score of 6.

---

> > > ### Author Response · Authors · 2026-04-04
> > >
> > > Thank you for your thoughtful follow-up and for your continued positive recognition of our paper. We are very glad that our response was able to address your concerns. We will incorporate them into the revised version to further improve the quality of the paper. We truly appreciate your time and constructive feedback!

---

### Official Review · Reviewer_1FRz · 2026-03-13

**Soundness:** 2
**Presentation:** 2
**Significance:** 3
**Originality:** 3
**Overall Recommendation:** 3
**Confidence:** 3

**Summary:**

The authors identify a critical challenge for large language models (LLMs) in multi-turn optimization: the trade-off between efficiency and effectiveness when learning from lightweight user feedback. Existing online reinforcement learning (RL) approaches can capture the dynamics of multi-turn interactions but incur high computational costs, whereas supervised fine-tuning (SFT) methods are efficient yet prone to distribution shift and behavioral collapse.

In this work, **DRIFT is proposed** as a novel framework that decouples trajectory sampling from policy optimization and leverages importance-weighted SFT to approximate a KL-regularized RL objective. DRIFT collects offline interaction trajectories from a fixed reference policy, computes return-based weights for each trajectory, and trains the policy via weighted SFT. Theoretical analysis shows that this weighted SFT is mathematically equivalent to the KL-regularized RL objective, thereby ensuring that the optimization aligns with the desired multi-turn behavior. Empirical results demonstrate that DRIFT matches or surpasses multi-turn RL baselines on both mathematical and general reasoning benchmarks, while maintaining training efficiency comparable to standard SFT methods.

**Compliance With Llm Reviewing Policy:**

Affirmed.

**Final Justification:**

The rebuttal usefully addressed several of my original concerns and improved my assessment of the paper. However, I still find the terminal-only design, the applicability boundary of the method, and parts of the presentation insufficiently resolved, so I maintain my original scores and Weak Reject recommendation.

**Key Questions For Authors:**

- How can DRIFT be scaled to multi-turn interactions with larger LLMs, such as qwen25-7b or qwen3 models?

- Beyond mathematical reasoning, how robust is DRIFT on open-ended multi-turn tasks?

- While DRIFT improves training stability, does it limit the ability to learn richer intermediate behaviors? What's the  main difference and advantage between DRIFT and DFT.

  **Reference:** On the Generalization of SFT: A Reinforcement Learning Perspective with Reward Rectification

**Limitations:**

This paper does not discuss Limitations, and please see weakness and question section.

**Strengths And Weaknesses:**

**Strengths:**

- **Technical soundness:** The derivation demonstrating the equivalence between KL-regularized RL and importance-weighted SFT is mathematically rigorous.
- **Practical significance:** DRIFT addresses efficient multi-turn optimization for LLMs, potentially enabling practical deployment without expensive online RL.
- **Novelty:** DRIFT introduces the decoupling of trajectory sampling from policy optimization and leverages importance-weighted SFT to approximate a KL-regularized RL objective, which is original.

**Weaknesses:**

- **Limited evaluation scope:** The paper lacks results on multi-turn dialogue datasets and on larger models such as Qwen2.5-7B and Qwen3-8B.
- **Complexity in presentation:** Some parts of the theoretical derivations are overly detailed and complex, which may hinder readability.
- **Open questions:** It is unclear whether DRIFT can be combined with online RL fine-tuning to further improve performance without significant computational overhead, and whether improving training stability might limit learning richer intermediate behaviors.
- Certain theorems or formula proofs are written too long and meaningless, further affecting readability.

---

> ### Author Rebuttal · Authors · 2026-03-31
>
> We sincerely thank the reviewers for recognizing the novelty, technical soundness, and practical significance of our work. We have carefully considered your questions and suggestions, provided our responses, and will incorporate them into the revised version. If you have additional concerns, please contact us for discussion.
> # W1 & Q1: Limited evaluation scope
> Thank you for this valuable question. DRIFT decouples rollout collection from optimization: trajectories are sampled once from a fixed reference policy, and training is then reduced to weighted SFT. As a result, its optimization phase is much closer to SFT than to online RL, making it naturally easier to scale to larger models than iterative online RL methods.
>
> To further address this concern, we additionally ran experiments on Qwen2.5-7B-Instruct. The detailed results are as follows:
>
> Method|MATH|MATH500|ThmQA|MMLU-R|MMLU-P|GPQA|All Avg
> ---|---:|---:|---:|---:|---:|---:|---:
> Base|62.2|64.0|37.1|90.5|68.9|66.1|64.8
> SFT-5turn|64.2|64.4|37.2|90.5|69.3|68.1|65.6
> UFO-5turn|66.3|66.2|38.6|92.3|70.6|73.2|67.9
> DRIFT-5turn|67.6|68.6|38.5|91.2|71.2|72.7|**68.3**
>
> # W2 & W4: Complexity in presentation
> Thank you for your valuable suggestion. In the revised version, we will strengthen the exposition of the reasoning process behind the theoretical content in Section 3 and improve the readability of this part.
>
> # W3: Open questions
> Thank you for your insightful question. Under the same number of training steps, we conducted additional experiments. The results show that continuing RL from DRIFT achieves better performance than continuing RL from SFT, and it also outperforms both pure RL and DRIFT alone. This demonstrates the potential of combining DRIFT with RL, either as a strong initialization for RL or as a way to further improve model performance. The detailed results are as follows.
>
> Method|MATH|MATH500|ThmQA|MMLU-R|MMLU-P|GPQA|Avg
> ---|---:|---:|---:|---:|---:|---:|---:
> UFO-5turn (200 steps)|55.5|56.4|33.5|87.0|57.1|71.2|60.2
> DRIFT-5turn (200 steps)|55.9|58.2|34.3|84.6|57.2|72.7|60.5
> SFT (100 steps) + UFO (100 steps)|58.3|57.4|34.5|82.1|55.6|62.1|58.3
> DRIFT (100 steps) + UFO (100 steps)|60.3|61.4|34.8|87.6|58.5|72.2|**62.5**
>
> # W3 & Q3: Richer intermediate behaviors
> Thank you for raising these valuable questions. In our implementation, DRIFT indeed retains only the terminal step for training, which may raise the concern that intermediate behaviors are not directly learned. However, under the setting considered in this paper, the behavior that truly needs to be learned is how to revise the response conditioned on the history of previous failed attempts. The training input still contains the full dialogue history, so DRIFT does not discard the context of intermediate behaviors; rather, it avoids treating failed content itself as the target to imitate. This is important because imitating incorrect responses can in fact hurt performance. As shown in Figure 6 of the appendix, retaining only the terminal step improves the final accuracy by 6.8%.
>
> In Appendix F.5, we also conduct an ablation on the repeated-answer penalty term. The results show that introducing this penalty term leads the model to explore more valid answers, indicating that DRIFT indeed enables the model to learn intermediate behaviors in the multi-turn error-correction process, such as avoiding repeated answers.
>
> # Q2: Open-ended multi-turn tasks
> Thank you for this insightful question. Our current study focuses on multi-turn correction under lightweight verifiable feedback, rather than fully open-ended dialogue. Following prior setups such as **SCoRe[1]** and **UFO[2]**, we use a deterministic verifier and fixed feedback to isolate the core correction-learning problem under verifiable rewards. Within this setting, DRIFT is not specific to mathematics: beyond math benchmarks, we also observe gains on broader general reasoning benchmarks, suggesting that DRIFT learns feedback-conditioned correction behavior rather than domain-specific patterns.Open-ended dialogue is beyond the current scope of this paper, but we agree that extending DRIFT to open-ended multi-turn tasks is an important direction for future work.
>
> [1] Training Language Models to Self-Correct via Reinforcement Learning
>
> [2] A Simple "Try Again" Can Elicit Multi-Turn LLM Reasoning
>
> # Q3: DRIFT vs DFT
> Thank you for the pointer. DFT improves SFT by correcting the token-level reward distortion induced by fixed expert supervision. DRIFT instead is derived directly from the KL-regularized multi-turn RL objective, yielding an offline trajectory-weighted SFT objective. Its main advantage is trajectory-level, return-aware credit assignment: DRIFT distinguishes early success, late success, and repetitive failures, making it more naturally suited to verifier-guided multi-turn correction than DFT.

---

> > ### Author Rebuttal · Reviewer_1FRz · 2026-04-04
> >
> > Thank you for the detailed rebuttal. Overall, the rebuttal addressed several of my original concerns and improved my evaluation of the paper. In particular, I appreciate the additional results on a larger backbone and the clarifications regarding the intended setting and the relation to prior methods. However, I still consider my concerns to be only partially resolved for three reasons.
> >
> > First, I remain somewhat unconvinced by the terminal-only training design. While the bias-variance argument is reasonable, I am still unsure whether useful signal from earlier turns is lost. More importantly, the current explanation suggests that the model mainly learns how to correct previous failed attempts, rather than how to reason step by step throughout the whole trajectory. I would appreciate a clearer explanation of whether this design improves the model’s underlying single-turn reasoning ability or mainly learns a correction policy.
> >
> > Second, I think the scope of the method should be stated more explicitly. Based on the rebuttal, the method seems most justified for short-horizon, verifier-guided tasks with lightweight and deterministic feedback. I encourage the authors to clarify this applicability boundary and discuss its limitations more explicitly.
> >
> > Third, I still find the presentation somewhat difficult to follow in parts. The revised paper would benefit from cleaner writing, a more concise theoretical presentation, and a clearer discussion of the practical motivation behind the design choices.
> >
> > Overall, the rebuttal improved my assessment, but these issues still prevent me from considering my concerns fully resolved

---

> > > ### Author Response · Authors · 2026-04-07
> > >
> > > Thank you very much for your careful follow-up and for recognizing parts of our rebuttal. Your remaining concerns are very helpful for more clearly positioning our method and improving the quality of the paper. We have carefully considered them and respond point by point below, hoping to address the concerns that remain.
> > > # 1. Explanation of the Terminal-Only Design
> > > Thank you for raising this very important clarification. Under the current setting of our paper, terminal-only supervision is more accurately understood as primarily learning a feedback-conditioned correction policy.
> > >
> > > In fact, terminal-only retains the full history as the conditioning context for the final step, what we discard is not the information contained in earlier turns as context, but rather the failed responses themselves as imitation targets. We cannot guarantee that all information from earlier turns is fully preserved. However, some useful information relevant to learning the correction policy does remain available, such as the answers from each previous turn in the context. Combined with the repeated-answer penalty, trajectories that repeat failed answers already present in the context receive lower weights within the same rollout set, which further encourages the model to avoid repeating ineffective answers and to attempt more effective revisions. This is consistent with our previous rebuttal. This effect should be understood as encouraging correction behavior across turns.
> > >
> > > Accordingly, we believe it is more appropriate to interpret this design choice in DRIFT as mainly improving the model’s ability to revise its answers based on prior failed attempts and lightweight feedback, rather than directly claiming that it systematically improves the model’s intrinsic single-turn reasoning ability in the absence of feedback.
> > >
> > > Our empirical results also support this interpretation. Figure 5 shows that DRIFT achieves a higher turn-by-turn correction rate than the baselines, which is more consistent with the interpretation that it learns a better correction policy. Figure 6 further shows that terminal-only training is superior to all-turn training in terms of training stability, final accuracy, and cumulative accuracy under every turn budget.
> > > # 2. Scope of the Method
> > > Thank you for this important suggestion. Clarifying the applicability boundary and limitations helps more precisely characterize the intended scope of the method. In its current form, DRIFT is best justified for short-horizon, verifier-guided multi-turn correction with lightweight deterministic feedback, which is also the setting assumed in our problem formulation and training protocol. Our current theoretical setup relies on a deterministic verifier and fixed feedback, and our empirical validation focuses on tasks with verifiable outcomes rather than fully open-ended dialogue. These settings follow the prior strong RL-based work we mentioned in our previous rebuttal, including **SCoRe [1]** and **UFO [2]**.
> > >
> > > We therefore do not intend the current paper to claim that DRIFT is a general solution for arbitrary multi-turn interaction. In particular, the present work does not directly validate settings with stochastic or ambiguous human feedback, open-ended conversational objectives, or substantially longer-horizon interactive planning. We will make these limitations explicit in the revised paper, rather than leaving them implicit.
> > >
> > > We will revise the relevant wording in the introduction and conclusion to more precisely characterize DRIFT’s intended scope as verifier-guided multi-turn correction, and we will add an explicit limitations discussion clarifying that our current evidence is strongest in short-horizon settings with deterministic feedback.
> > > # 3. Presentation
> > > Thank you for this very helpful suggestion. We apologize that, since the PDF cannot be revised during the rebuttal period, we are unable to directly present a fully revised manuscript at this stage. However, we will improve the presentation in the revised version in the following concrete ways:
> > >
> > > 1. We will reduce repeated phrasing and overly long or non-intuitive theoretical explanations. More generally, we will simplify the theoretical presentation to avoid unnecessary formalism, and make the discussion following each formula as concise and intuitive as possible, so that the role of each theoretical component is easier to understand.
> > > 2. We will simplify the proof chain in Section 3 by retaining only the core steps in the main text: from the optimal trajectory distribution, to connecting RL with distribution matching, and then to importance-weighted SFT. The remaining technical details and intermediate derivations will be moved to the appendix.
> > > 3. We will connect the theory and discussion of the terminal-only design more directly to Section 3, and also incorporate the above clarification regarding terminal-only training, so that the practical motivation behind our implementation becomes more natural and easier to follow.

---

### Official Review · Reviewer_dVqm · 2026-03-13

**Soundness:** 2
**Presentation:** 3
**Significance:** 3
**Originality:** 3
**Overall Recommendation:** 4
**Confidence:** 2

**Summary:**

(1) The paper develops DRIFT, an approach that decouples roll-out from optimization to realize a reinforcement learning objective via weighted SFT, which enables efficient training under multi-turn interaction protocols.

(2) The paper shows theoretical results which show that the weighted objective is mathematically equivalent to the underlying KL-regularized RL objective, ensuring the effectiveness and stability.

(3) With comprehensive experiments, the paper shows that DRIFT achieves performance similar to or better than online RL baselines across mathematical and general reasoning benchmarks and obtaining substantially higher training efficiency.

**Compliance With Llm Reviewing Policy:**

Affirmed.

**Final Justification:**

Thank you very much for the responses. I adjust the score accordingly.

**Key Questions For Authors:**

(also mentioned in the weaknesses)

(1) It would be great to add some justifications about the model choices.

(2) Related to the training efficiency, it would be great if the paper can discuss more analysis on different model backbones. It would also be interesting to discuss whether the speed up is still significant if we use more powerful GPUs (which are relatively easy to be accessed these days).

(3) Are the current results over multiple runs of experiments?

**Limitations:**

yes

**Strengths And Weaknesses:**

Strengths:

(1) The paper is well-motivated and addresses an important task.
(2) The paper provides comprehensive theoretical analysis and insights.
(3) The developed approach has good intuitions.

Weaknesses:
(1) The number of evaluated model backbones is limited. The used model backbones are not consistent with the model backbones used in the original paper of the baselines. For example, Qwen2.5-7B-Instruct is used in the UFO paper. But Qwen2.5-7B-Instruct is not used in the submitted paper (although Llama3.1-8B-Instruct is used). It would be great to add some justifications about the model choices.

(2) From the paper descriptions, it is not very clear what model is used to show the results in Figure 4. Considering that the introduction section highlights the contribution "offering substantially higher training efficiency.", it would be great if the paper can provide more analysis on different model backbones. It would also be interesting to discuss whether the speed up is still significant if we use more powerful GPUs (which are relatively easy to be accessed these days).

(3) Considering the accuracy performances of the developed approach and the baseline approaches are sometimes very similar, it would be very helpful to present the average results over multiple runs of experiments.

---

> ### Author Rebuttal · Authors · 2026-03-31
>
> We sincerely thank the reviewers for recognizing the value of our work, as well as its theoretical analysis and methodology. We have carefully considered your questions and suggestions, provided our responses, and will incorporate them into the revised version. If you have additional concerns, please contact us for discussion.
>
> # W1 & Q1: Model Choice
>
> Thank you for the careful observation. In the current submission, we evaluate DRIFT on two backbones: Qwen2.5-3B-Instruct and Llama3.1-8B-Instruct. We chose them to cover both different parameter scales and different model families, while keeping the comparison matrix tractable. Unlike UFO, where each backbone is compared with a smaller set of trained baselines, our paper evaluates each backbone against six trained baselines spanning offline vs. online and single-turn vs. multi-turn methods. Extending this full protocol to all five backbones used in UFO would require substantially more experiments.
>
> To further address your concern, we additionally conducted experiments on Qwen2.5-7B-Instruct. The detailed results are as follows:
>
> | Method      | MATH | MATH500 | ThmQA | MMLU-R | MMLU-P | GPQA |  All Avg |
> | ----------- | ---: | ------: | ----: | -----: | -----: | ---: | -------: |
> | Base        | 62.2 |    64.0 |  37.1 |   90.5 |   68.9 | 66.1 |     64.8 |
> | SFT-5turn   | 64.2 |    64.4 |  37.2 |   90.5 |   69.3 | 68.1 |     65.6 |
> | UFO-5turn   | 66.3 |    66.2 |  38.6 |   92.3 |   70.6 | 73.2 |     67.9 |
> | DRIFT-5turn | 67.6 |    68.6 |  38.5 |   91.2 |   71.2 | 72.7 | **68.3** |
>
>
>
> # W2 & Q2: Efficiency Results
>
> Thank you for the helpful suggestion. Figure 4 uses Qwen2.5-3B-Instruct, and we will clarify this in the revised version. To further cover different model backbones and GPU types, we additionally conducted training-efficiency experiments on Qwen2.5-3B-Instruct and Llama3.1-8B-Instruct using A800 80GB and H20 96GB GPUs. Following the same protocol as in the paper, we report training GPU time (hours), which is measured with a global batch size of 128 using 4 GPUs.
>
> The results follow the same trend as in Figure 4: as the number of turns increases, the training time of the RL method grows substantially, whereas the training time of DRIFT and SFT increases much more modestly and remains close to each other. The detailed results are as follows:
>
> **Qwen2.5-3B-Instruct**
>
> | GPU      | Turns |         SFT |       DRIFT |  UFO |
> | -------- | ----: | ----------: | ----------: | ---: |
> | A800 80G |     1 | 3.0 (40.0%) | 3.1 (38.0%) |  5.0 |
> |          |     3 | 3.3 (66.3%) | 3.6 (63.3%) |  9.8 |
> |          |     5 | 3.8 (73.2%) | 4.2 (70.4%) | 14.2 |
> | H20 96G  |     1 | 2.5 (35.9%) | 2.6 (33.3%) |  3.9 |
> |          |     3 | 3.0 (61.0%) | 3.2 (58.4%) |  7.7 |
> |          |     5 | 3.8 (69.4%) | 4.1 (66.9%) | 12.4 |
>
> **Llama3.1-8B-Instruct**
>
> | GPU      | Turns |         SFT |       DRIFT |  UFO |
> | -------- | ----: | ----------: | ----------: | ---: |
> | A800 80G |     1 |  6.1 (9.0%) |  6.1 (9.0%) |  6.7 |
> |          |     3 | 7.1 (34.3%) | 6.8 (37.0%) | 10.8 |
> |          |     5 | 8.2 (50.0%) | 8.4 (48.8%) | 16.4 |
> | H20 96G  |     1 | 5.8 (14.7%) | 6.0 (11.8%) |  6.8 |
> |          |     3 | 6.9 (35.5%) | 7.0 (34.6%) | 10.7 |
> |          |     5 | 8.8 (51.9%) | 9.1 (50.3%) | 18.3 |
>
> (Values in parentheses indicate the reduction relative to UFO.)
>
> # W3 & Q3: Multiple runs
>
> Thank you for your valuable suggestion. The results in the original submission are not averaged over multiple runs for two reasons. First, the baselines we follow, **SCoRe[1]** and **UFO[2]**, do not report multi-run statistics; in particular, SCoRe explicitly states that it trains only once and evaluates with greedy decoding, and we followed the same setup for fair comparison. Second, due to the large experimental cost, we adopted a fixed seed for training with greedy decoding for evaluation rather than repeated runs. However, for similar results, we agree that statistics over multiple runs can better demonstrate the effectiveness of DRIFT. We therefore conducted additional experiments on Qwen2.5-3B-Instruct using three different random seeds, focusing on comparisons with methods that yield very similar results, and report the mean and standard deviation to show that the observed gains are not due to chance. The detailed results are as follows:
>
> | Method      |       MATH |    MATH500 |      ThmQA |     MMLU-R |     MMLU-P |       GPQA |        Avg |
> | ----------- | ---------: | ---------: | ---------: | ---------: | ---------: | ---------: | ---------: |
> | UFO-5turn   | 55.6 ± 0.6 | 56.0 ± 0.3 | 33.7 ± 0.5 | 86.2 ± 0.9 | 57.2 ± 0.6 | 69.9 ± 2.1 | 59.8 ± 0.6 |
> | DRIFT-5turn | 56.0 ± 0.5 | 58.1 ± 0.4 | 34.2 ± 0.4 | 85.4 ± 1.0 | 57.2 ± 0.4 | 70.1 ± 2.4 | 60.2 ± 0.4 |
>
> [1] Training Language Models to Self-Correct via Reinforcement Learning
>
> [2] A Simple "Try Again" Can Elicit Multi-Turn LLM Reasoning

---

> > ### Author Rebuttal · Reviewer_dVqm · 2026-04-03
> >
> > Thank you very much for the responses. I would adjust the score.

---

> > > ### Author Response · Authors · 2026-04-04
> > >
> > > Thank you for your thoughtful follow-up and for raising the score. We are very pleased that the additional experiments and responses we provided were able to address your concerns, and that they also helped improve the quality of the paper. We will incorporate them into the revised version. We truly appreciate your time and constructive feedback!

---

### Official Review · Reviewer_qZUF · 2026-03-13

**Soundness:** 3
**Presentation:** 3
**Significance:** 3
**Originality:** 3
**Overall Recommendation:** 4
**Confidence:** 2

**Summary:**

For multi-turn RL settings, this paper starts from the perspective of decoupling rollout from policy optimization and shows that the KL-regularized RL objective is mathematically equivalent to importance-weighted supervised learning. It proves that the gradient of the online RL objective can be unbiasedly approximated by reweighting offline trajectories sampled from a fixed reference policy according to their exponentiated rewards. Based on this result, the paper proposes DRIFT. In the offline rollout stage, it collects interaction trajectories under a fixed reference policy and assigns return-based importance weights. In the trajectory-weighted SFT stage, it optimizes the policy with weighted SFT on the resulting dataset. Experimental results show that DRIFT matches or exceeds multi-turn reinforcement learning baselines while retaining the training efficiency of standard supervised fine-tuning.

**Compliance With Llm Reviewing Policy:**

Affirmed.

**Key Questions For Authors:**

See weaknesses.

**Limitations:**

yes

**Strengths And Weaknesses:**

### Strengths
1. The paper theoretically establishes the connection between the KL-regularized RL objective and importance-weighted supervised learning, which gives a solid foundation for the idea of decoupling rollout from optimization.
2. DRIFT is simple and practical, and compared with online RL it has a clear efficiency advantage while fitting the multi-turn setting well.
3. The paper includes sufficient ablation experiments and analysis, which provide good support for the design and empirical claims.


### Weaknesses
1. The equivalence from reverse-KL to forward-KL relies on a strong realizability assumption.
2. In Theorem 4, the derived objective is a trajectory-level objective, but the implementation only keeps Terminal-Step Retention. Although the paper provides a bias and variance analysis for the terminal approximation, its key conclusion is largely driven by the assumptions themselves. As a result, this is still a relatively loose bias-variance discussion, and it does not really show that the policy learned under terminal-only supervision is close to that under the original RL objective.

---

> ### Author Rebuttal · Authors · 2026-03-31
>
> We sincerely thank the reviewers for recognizing the theory, methodology, and empirical results of our work, as well as for their overall positive evaluation. We have carefully considered your questions and suggestions, provided our responses, and will incorporate them into the revised version. If you have any further concerns, please contact us for discussion.
>
> # W1: Realizability assumption
>
> Thank you for this important comment. We agree that our current Lemma 3 uses a realizability assumption to claim coincidence of the global minimizers of reverse KL and forward KL. More precisely, the exact statement in our theory is Theorem 2: the original KL-regularized RL objective is exactly equivalent to minimizing the reverse KL to $p^\star$. Realizability is only used when replacing this reverse-KL target by the forward-KL / weighted-SFT surrogate at the level of global optima.
>
> That said, we would like to stress that the non-realizable case still admits a meaningful local guarantee. For a fixed prompt $x$, when $p_\theta(\cdot\mid x)$ is close to $p^\star(\cdot\mid x)$, a Taylor expansion of both $\mathrm{KL}(p_\theta\|p^\star)$ and $\mathrm{KL}(p^\star\|p_\theta)$ around $p_\theta=p^\star$ shows that they have the same second-order term, so their mismatch is only higher-order small. Concretely, under a mild non-degeneracy condition on the relevant support, letting $\Delta_x:=\mathrm{TV}(p_\theta(\cdot\mid x),p^\star(\cdot\mid x))$, one obtains
> $$
> \mathrm{KL}(p_\theta\|p^\star)=\frac{1}{2}\sum_{\tau}\frac{(p_\theta(\tau\mid x)-p^\star(\tau\mid x))^2}{p^\star(\tau\mid x)}+O(\Delta_x^3),
> $$
>
> $$
> \mathrm{KL}(p^\star\|p_\theta)=\frac{1}{2}\sum_{\tau}\frac{(p_\theta(\tau\mid x)-p^\star(\tau\mid x))^2}{p^\star(\tau\mid x)}+O(\Delta_x^3),
> $$
>
> and therefore
> $$
> \big|\mathrm{KL}(p_\theta\|p^\star)-\mathrm{KL}(p^\star\|p_\theta)\big|\le C_x\Delta_x^3.
> $$
> Hence, even when $p^\star\notin\Pi_\theta$, the weighted-SFT objective remains a locally accurate surrogate for the original KL-regularized RL objective once $p_\theta$ is close to $p^\star$.
>
> We also agree that this local regime should be motivated rather than assumed without explanation. In our setting, it is induced by the KL-regularized objective itself: the optimizer satisfies
> $$
> p^\star(\tau\mid x)=\frac{1}{Z(x)}\,p_{\mathrm{ref}}(\tau\mid x)\exp(R(\tau)/\beta),
> $$
> so $p^\star$ is not an arbitrary target distribution but an exponential tilting of the reference policy. If $0\le R(\tau)\le R_{\max}$, then
> $$
> \mathrm{KL}(p^\star\|p_{\mathrm{ref}})\le R_{\max}/\beta,
> $$
> and thus $p^\star$ is constrained by design to remain near $p_{\mathrm{ref}}$ for moderate $\beta$. Since $p_\theta$ is initialized from $p_{\mathrm{ref}}$ and optimized under the same KL-regularized objective, the practically relevant regime is precisely one where both $p_\theta$ and $p^\star$ stay in a neighborhood of $p_{\mathrm{ref}}$, making $\Delta_x$ small. Therefore, while realizability is needed for an exact global equivalence claim, it is not needed for the practical local validity of our surrogate objective. We will include this clarification and the complete proof in the revised version.
>
> # W2: Terminal-only explanation
>
> Thank you for this insightful observation. We agree that Theorem 4 derives a full-trajectory objective, while terminal-step retention is a practical approximation rather than an exact implementation of that theorem. Our motivation is protocol-specific: under the stop-on-success setting, intermediate turns in a successful trajectory are precisely those after which the verifier still returned incorrect. Assigning the same large trajectory weight to those intermediate responses may therefore introduce misaligned credit assignment. In contrast, terminal-only supervision attaches the weight to the response that actually realizes success, while still conditioning on the full interaction history through $x_L$.
>
> Proposition 5 should be interpreted as a bias-variance motivation, not as a theorem showing closeness to the optimizer of the original RL objective. Our main support for terminal-step retention is therefore empirical. Not only performance comparable to the RL baseline. In Appendix E.1, we compare all-turn and terminal-only training using exactly the same offline trajectories and trajectory weights, differing only in whether intermediate turns are supervised. Terminal-only training is both more stable and achieves higher final accuracy:
>
> | Training Step         | 0    | 40   | 80   | 120  | 160  | 200  |
> | --------------------- | ---- | ---- | ---- | ---- | ---- | ---- |
> | All-turn Acc (%)      | 40.2 | 44.6 | 41.2 | 45.8 | 49.8 | 51.4 |
> | Terminal-turn Acc (%) | 40.2 | 49.6 | 55.0 | 53.4 | 54.0 | 58.2 |

---

> > ### Author Rebuttal · Reviewer_qZUF · 2026-04-03
> >
> > My main concerns are not fully resolved: the reverse-KL to forward-KL replacement still relies on a strong assumption, and there remains a gap between the trajectory-level objective in Theorem 4 and the terminal-only implementation.

---

> > > ### Author Response · Authors · 2026-04-07
> > >
> > > Thank you very much for your clear follow-up. We sincerely apologize for not having fully addressed your concerns. We have provided additional responses and discussion on your main remaining concerns, and we hope these clarifications will help resolve them while also improving the quality of the paper through this exchange.
> > >
> > > # 1. Realizability assumption
> > >
> > > We sincerely apologize that our previous response did not fully resolve your concern. We agree that the reverse-KL to forward-KL replacement in Lemma 3 relies on a strong realizability assumption. In our previous rebuttal, we discussed the case where realizability does not hold, and aimed to show that, in our setting, due to the constraint imposed by the KL-regularized RL objective, $p^\star$ and $p_{\mathrm{ref}}$ remain close, while $p_\theta$ starts from $p_{\mathrm{ref}}$ and moves toward $p^\star$ . This motivates a practically relevant local regime in which $p_\theta$ remains close to $p^\star$. Under this condition, $\Delta_x$ is small, and we can show that$\big|\mathrm{KL}(p_\theta\|p^\star)-\mathrm{KL}(p^\star\|p_\theta)\big|$ scales as $O(\Delta_x^3)$ . Therefore, reverse-KL and forward-KL are close, suggesting that even when the realizability assumption does not hold, although the exact mathematical equivalence is no longer available, using forward-KL in practice as an approximation to reverse-KL is still a reasonable choice.
> > >
> > > We also conducted a finite-support empirical comparison on Qwen2.5-3B-Instruct. Restricting both distributions to the same sampled trajectory support and renormalizing them there, **the estimated forward KL is about 0.79**, **the estimated reverse KL is about 0.71**, and their **absolute difference is about 0.08**. While this does not constitute a formal guarantee for the full trajectory space, it provides additional empirical evidence that, on the finite support relevant to our practical training setup, the mismatch between forward KL and reverse KL can be small in our setting.
> > >
> > > Therefore, what we would like to clarify is that, as you pointed out, claiming a mathematically equivalent reverse-KL to forward-KL replacement indeed requires the realizability assumption. However, when this assumption is not satisfied, we can still show that using forward-KL in practice as a surrogate for reverse-KL is reasonable. We will revise places in the paper that currently state mathematical equivalence, and make the wording more precise by presenting it instead as conditional equivalence or a practical approximation, in order to avoid misunderstanding. We will also add a discussion of the non-realizable case after Lemma 3, to better justify the practical choice.
> > >
> > > # 2. Terminal-only explanation
> > >
> > > Thank you for the follow-up. Theorem 4 yields a trajectory-level weighted objective, whereas our terminal-only retention is a protocol-specific design rather than an exact implementation of that theorem.
> > >
> > > Our point is not that terminal-only exactly recovers the optimizer of the full trajectory-level objective. Rather, under our stop-on-success protocol, terminal-only is a protocol-specific design used in the practical implementation. For a successful trajectory, every intermediate turn before the terminal one is a verifier-rejected failed attempt under fixed negative feedback. If the same large trajectory weight is applied to all of those turns, the update can retroactively assign positive credit to actions that did not realize success. Terminal-only supervision still conditions on the full interaction history, but concentrates the trajectory weight on the final response that actually realizes the verifier outcome.
> > >
> > > This is also the reason we framed Proposition 5 as a bias-variance motivation rather than an optimizer-closeness theorem. The role of terminal-only is to trade bounded bias for lower-variance, better-aligned credit assignment in this specific protocol.
> > >
> > > Importantly, our support is not only conceptual. In Appendix E.1, we compare all-turn and terminal-only using identical offline trajectories and identical trajectory weights, differing only in whether intermediate turns are supervised. Terminal-only is consistently more stable and reaches higher final accuracy. We will revise the paper to make this scope explicit that Theorem 4 provides the exact trajectory-level objective, while terminal-only is the practical optimization we use for stop-on-success multi-turn correction.

---

### Decision · Program_Chairs · 2026-04-30

**Decision:**

Accept (regular)

**Comment:**

DRIFT proposes a framework for efficient multi-turn LLM optimization by decoupling offline rollout collection from policy training, leveraging the theoretical equivalence between KL-regularized RL and importance-weighted SFT. Reviewer scores are 5 (ZhG2), 4 (qZUF), 4 (dVqm), and 3 (1FRz), with an overall lean toward weak accept despite flagged disagreement. The most confident reviewer, ZhG2 (confidence 4, score 5), found the theoretical analysis "nice and well-motivated," praised the clarity and rigor of experiments, and had their concerns fully resolved by the rebuttal. The two weak-accept reviewers (qZUF, dVqm; both confidence 2) acknowledged solid theory, practical efficiency advantages, and sufficient ablations, with dVqm fully satisfied after additional experiments on Qwen2.5-7B and multi-seed results. The main lingering concern from qZUF is that the reverse-KL to forward-KL replacement still rests on a realizability assumption, and the gap between the trajectory-level objective in Theorem 4 and the terminal-only implementation remains "relatively loose" in their view, though the authors provided reasonable local-validity arguments and empirical support.

The sole weak-reject, reviewer 1FRz (confidence 3), maintained their score citing three unresolved issues: the terminal-only design potentially discarding useful intermediate signal; the applicability boundary of DRIFT not being stated explicitly enough (best suited for short-horizon, verifier-guided settings with deterministic feedback); and presentation quality still "difficult to follow in parts." The authors acknowledged these concerns and committed to specific revisions, including moving intermediate derivations to the appendix and adding an explicit limitations section, which 1FRz recognized as improving their assessment without fully resolving it. I recommend Weak Accept, contingent on the authors following through on their promised presentation improvements and explicit scope/limitations discussion in the revision.